# BIRD-INTERACT: RE-IMAGINING TEXT-TO-SQL EVALUATION VIA LENS OF DYNAMIC INTERACTIONS

**Nan Huo**[α,γ]* **Xiaohan Xu**[α,γ]* **Jinyang Li**[α,γ]* **Per Jacobsson**[β] **Shipei Lin**[γ]
**Bowen Qin**[γ] **Binyuan Hui**[γ] **Xiaolong Li**[α,γ] **Ge Qu**[α,γ] **Shuzheng Si**[γ]
**Linheng Han**[γ] **Edward Alexander**[γ] **Xintong Zhu**[γ] **Rui Qin**[γ] **Ruihan Yu**[γ]
**Yiyao Jin**[γ] **Feige Zhou**[γ] **Weihao Zhong**[γ] **Yun Chen**[γ] **Hongyu Liu**[γ]
**Chenhao Ma**[γ†] **Fatma Ozcan**[β] **Yannis Papakonstantinou**[β] **Reynold Cheng**[α,γ†]

[α]The University of Hong Kong   [β]Google Cloud   [γ]The BIRD Team

bird.bench25@gmail.com
🎧 https://bird-interact.github.io

## ABSTRACT

Large language models (LLMs) have demonstrated remarkable performance on single-turn text-to-SQL tasks, but real-world database applications predominantly require multi-turn interactions to handle ambiguous queries, execution errors, and evolving user requirements. Existing multi-turn benchmarks fall short of capturing this complexity, either by treating conversation histories as static context or by limiting evaluation to narrow, read-only (SELECT-ONLY) operations, thereby potentially failing to reflect the challenges encountered in production-grade database assistant. In this work, we introduce BIRD-INTERACT, a benchmark that restores this missing realism through: (1) a *comprehensive interaction environment* that couples each database with a hierarchical knowledge base, metadata files, and a function-driven user simulator, enabling models to solicit clarifications, retrieve knowledge, and recover from execution errors without human supervision; (2) two *evaluation settings* reflecting real-world interaction settings which contain a pre-defined conversational protocol ($c$-Interact) and a more open-ended agentic setting ($a$-Interact) in which the model autonomously decides when to query the user simulator or explore the DB environment; (3) a *challenging task suite* that covers the full CRUD spectrum for both business-intelligence and operational use cases, guarded by executable test cases. Each task features ambiguous and follow-up sub-tasks, requiring LLMs to engage in dynamic interaction. The suite is organized into two sets: a full set (**BIRD-INTERACT-FULL**) of 600 tasks which unfold up to **11,796** dynamic interactions for a comprehensive overview of performance and a lite set (**BIRD-INTERACT-LITE**) of 300 tasks, with simplified databases for detailed behavioral analysis of interactions, and fast development of methods. Our empirical results highlight the difficulty of BIRD-INTERACT: the most recent flagship model **GPT-5** completes only **8.67%** of tasks in the $c$-Interact setting and **17.00%** in the $a$-Interact setting on the full task suite. Further analysis via memory grafting and Interaction Test-time Scaling (ITS) validates the importance of effective interaction for achieving success in dynamic text-to-SQL tasks.

## 1 INTRODUCTION

Data-driven decision-making has become indispensable across modern enterprises, prompting a surge of interest in Natural Language Interfaces to Databases (NLIDB) that empower non-technical users to extract insights from relational databases using natural language (Shi et al., 2024). Motivated by this vision, a wave of methods (Pourreza et al., 2025a;b; Pourreza & Rafiei, 2023; Liu et al., 2025; Qu et al., 2024; Li et al., 2025b; Maamari et al., 2024; Sheng & Shuai, 2025; Li et al., 2025a; Talaei

---

*Equal contribution.
†Corresponding authors are Reynold Cheng and Chenhao Ma.

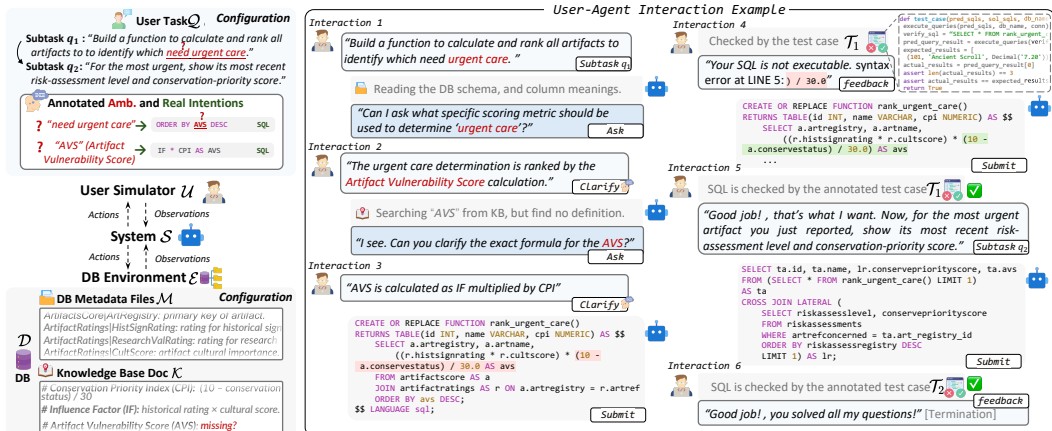

Figure 1: Task overview of BIRD-INTERACT showing the evaluated system interacting with DB Environment and User Simulator to complete the user task with a sequence of sub-tasks.

et al., 2024; Caferoğlu & Ulusoy, 2024; Cao et al., 2024; Lee et al., 2025) based on large language models (LLMs) has recently achieved impressive *text-to-SQL* performance on popular single-turn benchmarks such as Spider (Yu et al., 2018) and BIRD (Li et al., 2023b).

However, real-world data interaction is rarely a single, perfectly-formed query (Li et al., 2025c; Dinan et al., 2019). It is an iterative, stateful dialogue characterized by ambiguity (Chen et al., 2025b) and evolving goals (Wu et al., 2025). The task in Figure 1 exemplifies this complexity. To succeed, the text-to-SQL system must first engage the user to resolve the ambiguity of the term `urgent care`. Only with this clarified context can it generate the correct SQL. If its initial code fails an execution test, LLM must debug and revise its SQL solution based on the error feedback. After the user confirms the SQL is correct, they may proceed with a follow-up question that depends on its intermediate results. Therefore, evaluation on true practical utility LLMs with these multi-faceted aspects requires a benchmark containing a complete interactive problem-solving process, rather than isolated, single-turn SQL generation.

Although existing interactive text-to-SQL datasets (Yu et al., 2019b;a; Chen et al., 2025b; Guo et al., 2021; Dahl et al.) have been developed, they inadequately model this reality for two primary reasons. First, most multi-turn text-to-SQL benchmarks rely on **static conversation transcripts** (Yu et al., 2019a; Chen et al., 2025b; Yu et al., 2019b; Guo et al., 2021). They present models with a clean interaction history without recording the failed attempts, digressions, and clarifications that occur in practice. This design may introduce a fundamental limitation: every LLM is evaluated against the same predetermined dialogue trajectory, regardless of how it would have naturally guided the interaction. This setup fails to reward intelligent interaction strategies and cannot effectively penalize conversational mess up. Second, existing benchmarks suffer from a **narrow task scope**, focusing on read-only (`SELECT`-only) queries typical of business intelligence (BI) reporting. This ignores a vast and critical range of database management (DM) operations, including data manipulation (`INSERT`, `UPDATE`, `DELETE`), schema modifications (`ALTER TABLE`), and transactional control, which are also common operations in the normal DBA cycle (Chen et al., 2024).

To address these critical limitations, we introduce **BIRD-INTERACT**, a new benchmark designed to evaluate LLMs in a dynamic text-to-SQL environment. Our work makes the following contributions: **(1) A High-Fidelity Interactive Environment:** We develop a comprehensive sandbox upon an open-source project LIVESQLBENCH (BIRD-Team, 2025) for each task, including a hierarchical knowledge base (HKB) with domain-specific facts, metadata files, an executable database environment, and most critically, an interactive user simulator as recent research (Wu et al., 2025; Yao et al., 2025; Wang et al., 2024). This simulator can respond to clarification questions, provide feedback on proposed actions, and guide the model through complex tasks, enabling end-to-end evaluation without human intervention. However, recognizing that traditional simulators, even those powered by advanced models like `GPT-4o`, exhibit unfair behaviors such as ground-truth leakage, we propose a novel two-stage function-driven approach that maps model questions to constrained symbolic actions

before generating controlled simulator responses. **(2) Two Evaluation Settings:** We propose two popular evaluation settings. $c$-Interact (protocol-guided) presents tasks with a clear conversational protocol, testing a model's ability to follow a structured conversation with the user. In contrast, $a$-Interact (agentic) provides only a high-level goal, requiring the model to autonomously plan a strategy, decide when to query the database, consult documentation, or ask the user simulator for help. **(3) A Comprehensive and Challenging Task Suite:** BIRD-INTERACT expands the scope of evaluation to include the full spectrum of CRUD operations. Tasks are drawn from both analytical and operational domains and are accompanied by executable test cases that verify functional correctness. Each task features an ambiguous initial priority sub-task, dynamic clarification requirements, follow-up sub-tasks, and environmental uncertainties, which can only be resolved through dynamic interaction. The suite consists of two parts: a full set (**BIRD-INTERACT-FULL**) of 600 tasks, unfolding up to **11,796** dynamic interactions for a comprehensive evaluation of performance, and a lite set (**BIRD-INTERACT-LITE**) of 300 tasks with cleaner databases, enabling finer-grained behavioral analysis and faster deployment.

Our experiments show that state-of-the-art models struggle with BIRD-INTERACT, with **GPT-5** achieving only **8.67%** success in $c$-Interact and **17%** in $a$-Interact. We identify distinct challenges across interaction modes: communication effectiveness often determines success in $c$-Interact, while $a$-Interact suffers from bias toward costly trial-and-error over strategic resource exploration. We also observe Interaction Test-time Scaling (ITS), where performance improves monotonically with additional interaction opportunities across multiple models. These findings support our hypothesis that developing strategic interaction capabilities is key to improving LLM performance on complex database reasoning.

## 2 PROBLEM DEFINITION

**Task Definition.** We formalize interactive text-to-SQL as a multi-turn collaboration between a text-to-SQL system $\mathcal{S}_\theta$ and user simulator $\mathcal{U}_\gamma$ operating over database environment $\mathcal{E} = \{\mathcal{D}, \mathcal{M}, \mathcal{K}\}$, where $\mathcal{D}$ is the executable database, $\mathcal{M}$ contains schema metadata, and $\mathcal{K}$ represents external knowledge (Lee et al., 2021; Dou et al., 2022; Li et al., 2023b). Given a sequence of related sub-tasks $\mathcal{Q} = \{q_1, q_2, \ldots, q_n\}$, the goal is for $\mathcal{S}$ to generate SQL solutions $\{\sigma_1, \ldots, \sigma_n\}$ through interactions. For each sub-task $q_i$, the interaction proceeds through interaction turn $t = 1, 2, \ldots$ until completion:

$$u_i^t = \mathcal{U}_\gamma(h_i^{t-1}, q_i, \mathcal{E}), \quad s_i^t = \mathcal{S}_\theta(h_i^{t-1}, u_i^t, \mathcal{E}), \quad h_i^t = h_i^{t-1} \oplus \langle u_i^t, s_i^t \rangle \tag{1}$$

where $h_i^t$ represents the interaction history up to turn $t$ and $\oplus$ denotes text concatenation in prompt. The user simulator $\mathcal{U}_\gamma$ manages the interaction by presenting sub-tasks, answering clarification questions for ambiguous queries, and providing feedback on submitted SQL. Critically, subsequent sub-tasks are released only after successful completion of first sub-tasks.

**Metrics.** Each sub-task $q_i$ is annotated with ground-truth SQL $\sigma_i^*$ and executable test cases $\mathcal{T}_i$ that define correctness. A predicted solution $\sigma_i$ is correct if it passes all associated test cases, ensuring functional equivalence with $\sigma_i^*$. In our implementation, each task consists of two related sub-tasks ($n = 2$): an initial **priority sub-task** $q_1$ containing ambiguities requiring resolution, and (2) a subsequent **follow-up sub-task** $q_2$. We evaluate system performance using: (1) **Success Rate (SR)**: The proportion of sub-tasks completed successfully, with each sub-task scored 0 or 1. We report SR separately for sub-task 1 and sub-task 2 as an online evaluation during interaction. (2) **Normalized Reward**: Defined as normalized scoring according to priority weighting as designed in Appendix F to $[0, 1]$ for analyzing system behaviors after interaction (offline evaluation) (Yao et al., 2022).

## 3 BENCHMARK CONSTRUCTION

This section details the methodology for the construction of BIRD-INTERACT benchmark. We begin by outlining the overall benchmark setup (Section 3.1), and then elaborate on how we convert clear single-turn tasks into ones requiring interactions (Section 3.2).

### 3.1 SETUP AND RESOURCES

We build our benchmark on the text-to-SQL tasks and infrastructure of LIVESQLBENCH (BIRD-Team, 2025). We select this foundation due to several key advantages. First, LIVESQLBENCH provides a comprehensive evaluation environment. It supports the full spectrum of SQL operations, including DML and DDL, which allows for dynamic database states that reflect real-world usage. Furthermore, its permissive license and ready-to-use artifacts, including an executable database sandbox and metadata files, facilitate extension and reproducibility. Third, it features a Hierarchical Knowledge Base (HKB) that organizes external knowledge as nodes in a directed acyclic graph (DAG), as shown in Figure 1, where *"AVS"* depends on *"IF"* and *"CPI"*. This structure explicitly models dependencies between facts that require multi-hop reasoning to connect isolated information. Despite these strengths, LIVESQLBENCH is fundamentally a single-turn benchmark. This design fails to capture the interactive and often ambiguous nature of real-world data analysis scenarios. Our primary contribution is to convert this static benchmark into a dynamic, interactive setting.

### 3.2 INTERACTIVE TASK ANNOTATION

To maintain the integrity and quality of our benchmark, we recruit 12 expert annotators through a rigorous multi-stage selection process detailed in Appendix C. We describe systematically the conversion from single-turn tasks of LIVESQLBENCH into multi-turn interactive scenarios through two key annotation strategies: ambiguity injection and follow-up sub-task generation:

**Ambiguity Injection.**    Ambiguities in daily life require interactions to seek clarification. To make annotation and evaluation controllable, we design methods to inject ambiguities into single-turn queries and the environment from LIVESQLBENCH, pairing each with a unique clarification.

**(1) Superficial user query ambiguities**: we target surface-level ambiguity in the user request. These include *intent-level ambiguities*, where the user language is vague (e.g., `"elderly people"`), and *implementation-level ambiguities*, where the user's intent is clear but the implementation details (e.g., decimal precision) are under-specific. **(2) Knowledge ambiguities:** we inject incompleteness into the external knowledge. This category includes two subtypes: (i) *one-shot knowledge ambiguity*, where isolated knowledge entries are removed. (ii) *knowledge chain breaking*, where intermediate nodes in multi-hop knowledge chains are masked. For example, consider the chain `"urgent care"` → `"AVS"` → `"IF/CPI"` in Figure 2. By masking the intermediate node, i.e., the fact `"AVS"` in HKB, we deliberately break the inferential chain, rendering knowledge ambiguous and requiring user clarification to proceed. **(3) Environmental ambiguities:** LIVESQLBENCH databases already contain natural noise, such as NULL in critical fields, which further introduces uncertainty in how these cases should be handled.

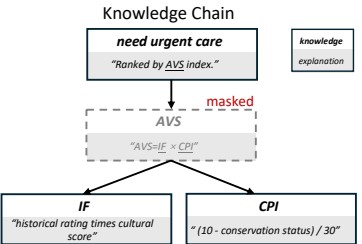

Figure 2: Knowledge chain breaking ambiguity.

Each injected ambiguity is paired with a corresponding SQL snippet from the ground-truth query as a *clarification source*, which guides our user simulator in generating consistent and contextually appropriate clarifications. Quality control ensures that ambiguous queries are unsolvable without clarification yet fully reconstructable once clarifications are provided. Complete details are given in Appendix H.

**Follow-Up Sub-tasks Annotation.**    User intents frequently evolve throughout an interactive session (Taylor, 2015), with users modifying, filtering conditions, or exploring related aspects of their queries. Therefore, we also extend each initial priority sub-task with one additional follow-up sub-task to resonate with this scenario.

These follow-up sub-tasks are designed carefully using a principled 5-category taxonomy detailed in Appendix H.5. A key contribution of our benchmark is the introduction of **state dependency** between sub-tasks, different with other datasets (Yu et al., 2019a;b; Lee et al., 2021; Zhong et al., 2017; Li et al., 2025d). System models must reason over modified database states or the newly created objects (e.g. tables) from preceding queries to write SQLs for follow-up sub-tasks.

### 3.3 FUNCTION-DRIVEN USER SIMULATOR

Evaluating interactive text-to-SQL systems requires user interactions, such as multi-turn requests and responses to clarification questions. Conducting such human-in-the-loop evaluations at scale is impractical. To make large-scale evaluations feasible, recent interactive benchmarks, such as MINT (Wang et al., 2024), employ LLMs to simulate human users (Li et al., 2025c; Yu et al., 2019a;b). However, we observe that there are two major issues among these simulators: (1) they sometimes leak information from ground-truth SQL query, and (2) they may deviate from the original task requirements (Barres et al., 2025; Kazi et al., 2024).

**Two-Stage Strategy.** To ensure a more robust evaluation, we introduce a two-stage **function-driven user simulator**, as illustrated in Figure 3(c). In the first stage, an LLM functions as a semantic parser. It maps the system's clarification request into one of three predefined allowed actions: `AMB()`, `LOC()`, or `UNA()`. `AMB()` is invoked for queries related to ambiguities that have been pre-annotated with the key SQL snippet. `LOC()` handles reasonable clarification requests that fall outside our pre-annotated ambiguities, such as questions about SQL formatting or specific sub-components. In these cases, the simulator uses an AST-based retrieval step to locate the relevant SQL fragment (detailed in Appendix N). Finally, `UNA()` rejects any inappropriate requests, such as attempts to elicit ground-truth answers. In the second stage, the user simulator generates a final response based on the chosen action and the annotated GT SQL with clarification source. This two-stage approach ensures the simulator's behavior remains predictable and controllable, while still permitting diverse and context-aware interactions. Detailed prompts are provided in Appendix R.

Table 1: Data Statistics

| STATISTIC | LITE | FULL |
|---|---|---|
| **Total Tasks** | 300 | 600 |
| # BI tasks | 195 | 410 |
| # DM tasks | 105 | 190 |
| # Distinct Test Cases | 135 | 191 |
| # Tokens / User Query | 40.22 | 32.95 |
| # Tokens / SQL | 361.52 | 252.21 |
| # Ambiguities / Task | 5.16 | 3.89 |
| # sub-tasks / Task | 2 | 2 |
| # Interactions / Task | 13.04 | 13.64 |
| Inter-Agreement | 93.33 | 93.50 |

### 3.4 DATA STATISTICS

Table 1 reports key properties of BIRD-INTERACT. The resulting benchmark comprises a total of 900 interactive text-to-SQL tasks, each featuring an ambiguous initial priority sub-task, dynamic clarification requirements, follow-up sub-tasks, and environmental uncertainties, collectively spanning the full CRUD spectrum (Create, Read, Update, and Delete). In Appendix E, we also conduct a comprehensive comparison against other relevant benchmarks, showing that BIRD-INTERACT is among the most open, challenging, and long-horizon interactive benchmarks in text-to-SQL scenarios.

## 4 EVALUATION SETTINGS

**Two Evaluation Settings.** The interactive framework of BIRD-INTERACT supports evaluation in two scenarios: LLMs as conversational assistants (*c*-**Interact**) (Dinan et al., 2019) and as agents (*a*-**Interact**) (Schluntz & Zhang, 2024).

**Budget-Constrained Awareness Testing.** The application of LLMs is limited by computational resources and user patience (Wen et al., 2025; Li et al., 2025e). We introduce a **budget-constrained awareness** mechanism to both evaluation settings, where interactions are capped by an adaptive budget and systems are informed of the remaining budget. This enables evaluation under varying budgets, including **stress-testing** (Ahmad et al., 2025; Zhang et al., 2025) in low-budget conditions to assess the system's ability to ask the right questions and plan effectively. The specific budget settings are detailed in the following sections.

### 4.1 *c*-INTERACT EVALUATION

**Interaction Setup.** The *c*-Interact evaluation establishes a multi-turn dialogue between user simulator $\mathcal{U}$ and system $\mathcal{S}$. The session unfolds in two sequential phases of sub-tasks: First, $\mathcal{U}$ presents an underspecified sub-task $q_1$ alongside database metadata $\mathcal{M}$ and knowledge base $\mathcal{K}$. System $\mathcal{S}$ may engage in clarification dialogue before generating SQL $\sigma_1$. Upon successful validation against test cases $\mathcal{T}_1$, $\mathcal{U}$ issues a contextually coherent follow-up sub-task $q_2$, prompting $\mathcal{S}$ to respond with SQL $\sigma_2$. Each sub-task incorporates a single debugging opportunity: following query failure, $\mathcal{S}$ may

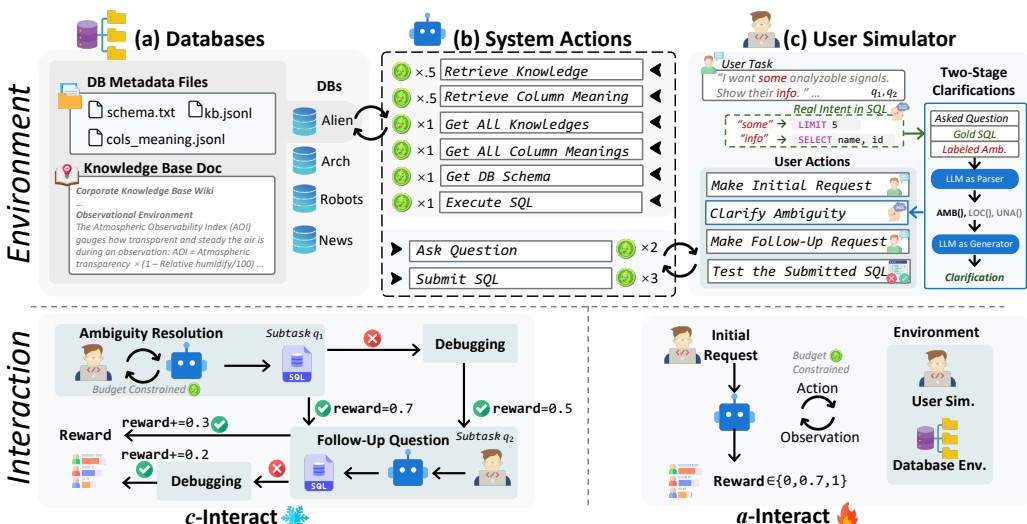

Figure 3: Two evaluation settings for BIRD-INTERACT: $c$-Interact, where the system engages in conversation with the user, and $a$-Interact, where the system interacts flexibly. At the end of the task, the system will receive a reward $r \in [0, 1]$.

submit one revised query after receiving execution feedback from $\mathcal{U}$. Each debugging attempt incurs a reward penalty to account for the additional computational cost, details can be found in Figure 3. The evaluation episode concludes when both sub-tasks are successfully completed or all attempts are exhausted. Notably, failure in the initial priority sub-task immediately terminates the entire session.

**Budget Constraints.** The budget is implemented as a constraint on the number of clarification turns. The total allowed turns, $\tau_{\text{clar}}$, are calculated as follows: $\tau_{\text{clar}} = m_{\text{amb}} + \lambda_{\text{pat}}$.

Here, $m_{\text{amb}}$ represents the minimum budget required to resolve the ambiguities, which is equal to the number of annotated ambiguities in the user task. The parameter $\lambda_{\text{pat}}$ is a tunable variable that simulates different levels of user patience, granting the evaluated system extra turns for clarification.

## 4.2 $a$-INTERACT EVALUATION

**Interaction Setup.** The $a$-Interact provides LLMs with autonomous planning and execution within a pre-defined action space, following REACT paradigm (Yao et al., 2023). We model the complete database environment as a set of callable tools, containing the target database, metadata, HKB, and User Simulator, allowing the agent to determine optimal invocation strategies dynamically. In this work, we summarize and define 9 discrete actions common to text-to-SQL with details in Appendix J. BIRD-INTERACT also supports customized scaffolds, details can be found in Appendix J.2.

**Budget Constraints.** To reflect the varying computational costs of different actions, we implement a budget-constrained evaluation framework where each action consumes a predetermined amount of budget, encouraging cost-effective action sequences. The total budget for each task is $B = B_{\text{base}} + 2\,m_{\text{amb}} + 2\,\lambda_{\text{pat}}$, where $B_{\text{base}} = 6$ is the base budget, $m_{\text{amb}}$ is the number of annotated ambiguity points, and $\lambda_{\text{pat}}$ is the user patience parameter, maintaining consistency with the $c$-Interact framework. This setting evaluates the agent's ability to achieve high performance under resource constraints while balancing thoroughness with efficiency. Further details of action costs are provided in Appendix J.

This setting can evaluate agent performance under realistic constraints that present practical database interaction scenarios, where users have limited patience and computational resources are finite.

Table 2: Success Rate and Final Normalized Reward of different models on BIRD-INTERACT-FULL. The success rate is cumulative; Reward* is the normalized reward. The values reported in $c$-Interact are after debugging phase, and (+n) means the performance gained via debugging. Avg. Cost is the cost for one task on average in USD. Our user simulator has an avg. cost of 0.03 USD. BI = Business Intelligence User Queries, DM = Data Management User Queries.

| Model | Priority Question (Success Rate %) ↑ | | | Follow Ups (Success Rate %) ↑ | | | Reward* ↑ | Avg. Cost ↓ |
|---|---|---|---|---|---|---|---|---|
| | BI | DM | Overall | BI | DM | Overall | | |
| *c-Interact Text-to-SQL* | | | | | | | | |
| GPT-5 | 9.49 (+0.00) | 25.40 (+2.12) | 14.50 (+0.67) | 5.84 (+0.24) | 14.81 (+0.53) | 8.67 (+0.33) | 12.58 | $ 0.08 |
| Claude-Sonnet-3.7 | 10.71 (+4.62) | 33.86 (+7.41) | 18.00 (+5.50) | 4.62 (+0.49) | 16.40 (+3.17) | 8.33 (+1.33) | 13.87 | $ 0.29 |
| Deepseek-Chat-V3.1 | 11.44 (+0.73) | 33.86 (+3.17) | 18.50 (+1.50) | 4.62 (+0.24) | 16.93 (+1.06) | 8.50 (+0.50) | 15.15 | $ 0.12 |
| Qwen-3-Coder-480B | 16.30 (+2.68) | 34.39 (+5.29) | 22.00 (+3.50) | 8.03 (+0.97) | 16.93 (+4.23) | 10.83 (+2.00) | 17.75 | $ 0.11 |
| Claude-Sonnet-4 | 16.06 (+4.87) | 35.98 (+10.58) | 22.33 (+6.67) | 10.46 (+1.22) | 22.22 (+3.70) | 14.17 (+2.00) | 18.35 | $ 0.29 |
| O3-Mini | 17.76 (+2.92) | 37.57 (+11.11) | 24.00 (+5.50) | 11.44 (+0.73) | 25.40 (+4.23) | 15.83 (+1.83) | 20.27 | $ 0.07 |
| Gemini-2.5-Pro | 18.73 (+4.38) | 38.62 (+10.05) | 25.00 (+6.17) | 12.41 (+1.22) | 24.87 (+5.29) | 16.33 (+2.50) | 20.92 | $ 0.04 |
| *a-Interact Text-to-SQL* | | | | | | | | |
| Qwen-3-Coder-480B | 8.05 | 24.74 | 13.33 | 3.90 | 4.74 | 4.17 | 10.58 | $ 0.07 |
| Deepseek-Chat-V3.1 | 10.49 | 31.58 | 17.17 | 4.63 | 5.26 | 4.83 | 13.47 | $ 0.06 |
| O3-Mini | 12.20 | 36.32 | 19.83 | 5.85 | 14.21 | 8.50 | 16.43 | $ 0.06 |
| Gemini-2.5-Pro | 10.49 | 41.58 | 20.33 | 5.85 | 20.00 | 10.33 | 17.33 | $ 0.22 |
| Claude-Sonnet-3.7 | 11.46 | 41.58 | 21.00 | 5.61 | 16.84 | 9.17 | 17.45 | $ 0.60 |
| Claude-Sonnet-4 | 15.85 | 53.68 | 27.83 | 8.05 | 22.63 | 12.67 | 23.28 | $ 0.51 |
| GPT-5 | 15.61 | 58.42 | 29.17 | 10.98 | 30.00 | 17.00 | 25.52 | $ 0.24 |

## 5 EXPERIMENT

We benchmark 7 recent and powerful LLMs (2 open-source, 5 closed-source) as system models via a fresh PostgreSQL 14 Docker instance for more stable evaluation. We set the user patience to 3 by default and $a$-Interact base budget of 6. All models use temperature=0 and top_p=1, with default reasoning settings, conducting single runs due to cost (full details in Appendix I.2 and I.3).

### 5.1 MAIN RESULTS

Table 2 summarizes the success rate (SR) and normalized reward (NR) obtained by 7 representative frontier LLMs on BIRD-INTERACT-FULL. The full experimental results of BIRD-INTERACT-LITE can be found in Table 10. We can observe:

**BIRD-INTERACT Remains Challenging, Leaving Ample Room for Future Improvement.** Even the strongest models in our study, `Gemini-2.5-Pro` and `GPT-5`, capture only 20.92% and 25.52% of the available reward respectively, in the $c$-Interact and $a$-Interact mode. Absolute success rates reveal similar limitations: no more than 16.33% of tasks are solved end-to-end in $c$-Interact and 17.00% in $a$-Interact, with most models falling in substantially lower rates.

**Evolving User Intent is a Challenge in Online Assessment.** Follow-up sub-tasks are noticeably more challenging, likely because the longer, concatenated context in these turns remains a bottleneck for LLMs in interactive text-to-SQL tasks.

**Offline Reward v.s. Online SR Evaluation.** Table 2 shows that offline normalized reward (NR) and online success rate (SR) generally correlate positively, though notable divergences occur due to the reward structure allocating 70% to the primary sub-task and 30% to follow-up sub-tasks. These complementary metrics capture different aspects of model performance. Success rate measures holistic task completion across multi-turn interactions, relevant when users prioritize successful outcomes regardless of path. Normalized reward assesses performance on users' critical initial objectives while crediting challenging follow-up sub-tasks. Together, they provide comprehensive evaluation of the distinct capabilities required for advanced interactive text-to-SQL systems.

**Business Intelligence versus Data Management.** Business intelligence (BI) queries pose significantly greater challenges for LLMs compared to data management (DM) tasks since DM operations typically follow standardized, predictable patterns that LLMs can effectively learn (Li et al., 2025d),

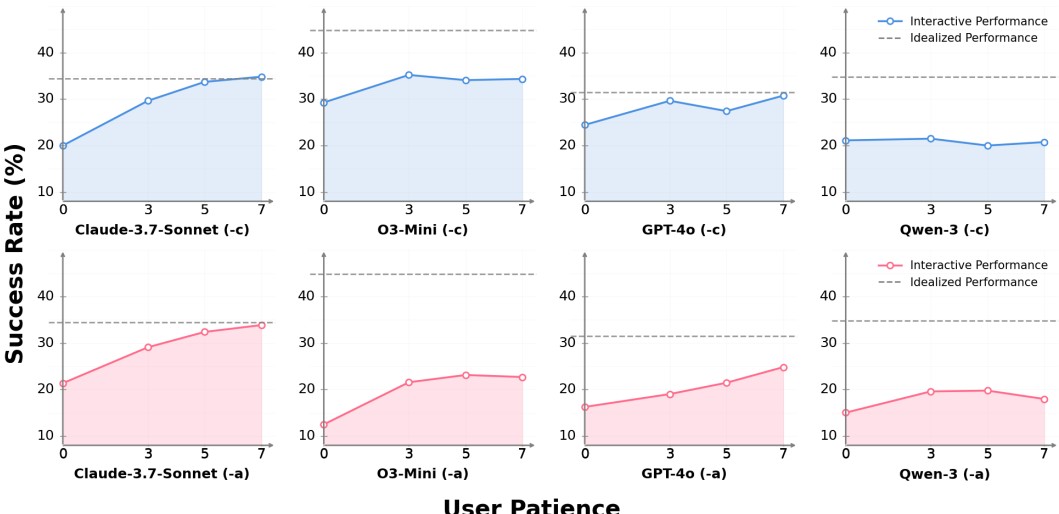

Figure 4: The performance of different LLMs with different user patience on BIRD-INTERACT-LITE. The red line denotes $a$-Interact mode (-a); the blue line denotes $c$-Interact mode (-c). And the dotted line (***Idealized*** Performance) denotes the performance under ambiguity-free single-turn text-to-SQL.

whereas BI queries demand nuanced understanding of complex, domain-specific business logic and analytical reasoning that varies substantially across contexts.

**Interaction Mode Emerged as the Decisive Factor for a Successful Outcome.** Furthermore, we observe that different models demonstrate varying aptitudes for different interaction paradigms, with each model showing relative strengths in specific modes. For example, `GPT-5` performs poorly in the constrained, predefined flow designed personally of the $c$-Interact mode by achieving only 14.50% SR (worst) but excels in the $a$-Interact setting with 29.17% SR (best), which affords more flexible and exploratory space. This evidence demonstrates the critical importance of matching interaction modes to model-specific capabilities, which we hypothesize stem from differences in training data distributions and architectural inductive biases (Liu et al., 2024; Gao et al., 2024b).

## 5.2 INTERACTION ANALYSIS

**The Impact of Communication on Task Success in $c$-Interact.** A notable finding is the underperformance of the flagship model, `GPT-5`, on the $c$-Interact, despite its strong performance on many single-turn tasks (Phan et al., 2025; Glazer et al., 2024; Rein et al., 2024). Therefore, we hypothesize that this stems from a deficiency in its interactive communication abilities rather than its core generation capability. To test this hypothesis, we conduct an experiment termed ***Memory Grafting***. In this setup, we provide `GPT-5` with the ambiguity resolution histories from **two other better models**, `Qwen-3-Coder` and `O3-mini`, before asking it to generate the final SQL query. The results, presented in Figure 5, show that `GPT-5`'s performance improves significantly when leveraging the interaction history from either model. This finding indicates that while `GPT-5` possesses robust SQL generation capabilities, a more effective communication schema is required to help it achieve satisfactory outcomes for user tasks. We also further analyze the patterns for effective communication in Appendix P.

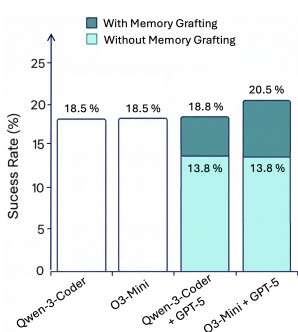

Figure 5: SR of GPT-5 with memory grafting.

**Interaction Test-Time Scaling.** To investigate the relationship between interaction frequency and model performance, we conduct an Interaction Test-Time Scaling (ITS) experiment in **BIRD-INTERACT-LITE** where results are shown in Figure 4. We simulate varying levels of user patience by allowing different numbers of interaction turns for both $c$-Interact and $a$-Interact. As a baseline, we include single-turn task performance for each model, where all necessary context is provided to

create unambiguous tasks. This single-turn condition represents an ***idealized*** scenario that, while potentially requiring significant user effort to ensure complete information provided (Li et al., 2025d), eliminates the need for further clarification. As demonstrated in the figure, `Claude-3.7-Sonnet` exhibits clear scaling behavior with respect to increasing interaction opportunities.

This pattern shows that the model can steadily improve by transforming additional interaction chances into valuable information gains through efficient interaction.

> **ITS Law:** A model satisfies this law if, given enough interactive turns, its performance can match or even surpass that of the idealized single-turn task.

**Action Distribution Patterns in $a$-Interact.** We analyze action distributions across 7 system models and find concentration in two primary actions: `submit` (direct code execution with error feedback) and `ask` (user clarification requests), which together comprise 60.87% of all actions. Despite being the most computationally expensive actions (Figure 3), models favor these over systematic exploration behaviors like knowledge and schema retrieval. This suggests LLMs prefer direct trial-and-error execution over comprehensive environment exploration, likely due to pre-training biases. Future work should incentivize broader tool utilization for complex interactive tasks. Additional analysis on the FULL set appears in Appendix J.

## 6 USER SIMULATOR ANALYSIS

This section presents a comprehensive evaluation of our function-driven user simulator compared to conventional user simulators and their respective impacts on dynamic interactive text-to-SQL benchmarks through both objective and subjective experiments.

**Evaluation on USERSIM-GUARD.** To provide an objective and comprehensive observation of different user simulator mechanisms, we construct a static dataset called USERSIM-GUARD, comprising 2,100 questions with reference actions labeled by human experts. Detailed information regarding the distribution and annotation procedures can be found in Appendix O. We employed an LLM-as-Judge (Zheng et al., 2023) evaluation framework using `Qwen3-235B-A22B-Instruct-2507` as independent evaluators. Our analysis reveals significant reliability concerns with conventional user simulator designs. Specifically, as shown in Figure 6, when confronted with Unanswerable (UNA) questions, baseline user simulators

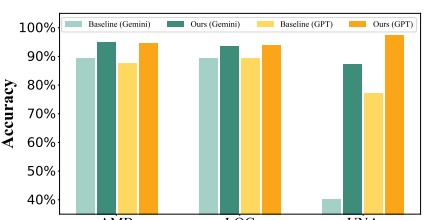

Figure 6: The accuracy of different user simulators on USERSIM-GUARD.

consistently fail to implement safeguards, resulting in unfair or inappropriate feedback generation with failure rates reaching up to 67.4% depending on the backbone model. In contrast, our proposed function-driven approach demonstrates substantially improved reliability, reducing the failure rate to as low as 2.7%. This represents a dramatic improvement in user simulator robustness and reliability compared to baseline approaches.

**Alignment with Human User.** We evaluate alignment between our user simulators and actual human behavior by having human experts interact with 7 system models on 100 randomly sampled tasks across BI and DM domains. We then compute correlations (Ivey et al., 2024; Kong et al., 2024) between success rates (SR) achieved by human users versus our simulators across the same tasks. As shown in Table 3, function-driven simulators demonstrate significantly stronger alignment with human behavior: `GPT-4o` with function calling achieves **0.84** Pearson correlation (p = 0.02) compared to 0.61 without function calling (p = 0.14), while Gemini-2.0-Flash shows similar improvements.

Table 3: Correlation analysis between AI and human users.

| User Simulator | Pearson (*p*-value) |
|---|---|
| GPT-4o 
 - w/ Func. (Ours) | 0.84 (p = 0.02) |
| Gemini-2.0-Flash 
 - w/ Func. (Ours) | 0.79 (p = 0.03) |
| GPT-4o 
 - Baseline | 0.61 (p = 0.14) |
| Gemini-2.0-Flash 
 - Baseline | 0.54 (p = 0.21) |

These results confirm that incorporating our designed mechanism produces more realistic user simulators that better reflect actual human-AI interaction patterns (detailed analysis in Appendix O).

## 7 RELATED WORK

**Text-to-SQL.** Text-to-SQL has emerged as an attractive interface to relational databases because it frees users from learning intricate schema details and SQL syntax. The advent of large language models (LLMs) (OpenAI, 2025; Team et al., 2023; Team, 2024; Guo et al., 2025; Li et al., 2023a; Qu et al., 2025) with strong reasoning and cross-domain generalization has accelerated this progress. Few-shot systems such as DIN-SQL (Pourreza & Rafiei, 2023) and DAIL-SQL (Gao et al., 2024a) exploit in-context learning to decouple the task into schema-linking and SQL-generation stages, while methods like CodeS (Li et al., 2024a) and DTS-SQL (Pourreza & Rafiei, 2024) improve smaller models through carefully curated, high-quality training subsets. Concurrently, agent-based frameworks that interleave thought, action, and observation, which are exemplified by MAC-SQL (Wang et al., 2025), demonstrate that iterative interaction with the environment can further raise SQL accuracy. Despite these advances, virtually all existing systems are evaluated only in single-turn settings; their effectiveness in conversational, multi-turn text-to-SQL scenarios remains an open question.

**Multi-turn Text-to-SQL.** Multi-turn Text-to-SQL addresses the reality that user queries are often ambiguous or underspecified; without clarification the system may return incorrect or empty results. Benchmarks such as CoSQL and LEARN-TO-CLARIFY extend the Spider (Yu et al., 2018) dataset with dialogue turns to probe this challenge (Yu et al., 2019a; Chen et al., 2025b; Li et al., 2024b). However, these resources presuppose a static, noise-free dialogue history shared by all models, ignoring that different systems might ask different follow-up questions (Yao et al., 2025; Barres et al., 2025). More recent evaluations of autonomous agents, for example, MINT, introduce dynamic interaction histories (Wang et al., 2024), yet they have not been adapted to the text-to-SQL setting. Constructing a realistic user simulator for databases is non-trivial because it must respect complex schema constraints while keeping the answer space fair and controllable (Zhou et al., 2025; Barres et al., 2025). In this work, we fill this gap by proposing an interactive benchmark that is implemented with an optimized user simulator, new databases, and knowledge, and we analyze the behaviour of state-of-the-art reasoning models rigorously to make contributions for realistic and uncertain text-to-SQL systems.

## 8 FUTURE WORK

While BIRD-INTERACT establishes a comprehensive framework for evaluating interactive text-to-SQL systems, several directions remain for future investigation. First, we plan to develop a post-trained, human-aligned local user simulator via post-training, aiming to capture more reliable response patterns while maintaining controllability and reducing API cost. Second, our current $a$-Interact setting imposes strict budget constraints that create a **stress-mode** evaluation environment, placing considerable pressure on LLM agents to make optimal decisions under resource scarcity. To complement these findings, we will conduct experiments in a **free-mode** setting without the budget-constrained awareness testing (Section 4). This could allow us to observe natural interaction strategies when models are unconstrained, identify whether more sophisticated exploration patterns emerge, and characterize the relationship between interaction thoroughness and task success. Comparing stress-mode and free-mode performance will provide deeper insights into efficiency-effectiveness trade-offs in interactive text-to-SQL systems.

## 9 CONCLUSION

We present BIRD-INTERACT, a benchmark for evaluating interactive text-to-SQL systems through dynamic, multi-turn interactions that better reflect real-world usage scenarios. Our benchmark features a function-driven user simulator, dual evaluation settings for conversational and autonomous planning modes, and totally 900 challenging tasks designed to test LLM abilities to handle ambiguities and maintain state across turns. Comprehensive evaluation demonstrates a critical gap between existing SQL generation capabilities and the strategic interaction skills required for effective human-AI collaboration in database querying.

## ACKNOWLEDGEMENT

Reynold Cheng, Nan Huo, Xiaohan Xu, Jinyang Li, Xiaolong Li, and Ge Qu were supported by the Research Grant Council of Hong Kong (RGC Project HKU 17202325), the University of Hong Kong (Project 2409100399), and the HKU Faculty Exchange Award 2024 (Faculty of Engineering). And we would like to express our sincere gratitude to Irina Saparina, Mohammadreza Pourreza, Mehdi Bouzouina, Hailong Li, Jiatong Shi, and Professor Shinji Watanabe for their fruitful discussions and valuable insights that helped improve this work.

## ETHICS STATEMENT

This research complies with the ICLR Code of Ethics. We have carefully reviewed the guidelines and ensured that our work aligns with the stated ethical standards, including considerations of data privacy, fairness, and responsible use of released datasets and code. Justification: This work does not involve crowdsourcing or research with human subjects. All annotation and task creation were carried out by the authors themselves.

## REPRODUCIBILITY STATEMENT

We have taken several steps to ensure the reproducibility of our work. First, for the evaluated systems, all experimental settings, including model parameters, temperature ($= 0$), and budget configurations, are clearly documented in Appendix I, ensuring that our evaluation can be replicated under the same conditions. Each task is executed in a freshly re-initialized PostgreSQL 14 instance (Docker). The Docker image contains the database engine and benchmark code, and is restarted for every run to guarantee a clean and consistent state. This setup makes experiments deterministic, isolated across runs, and easy to reproduce by rebuilding the environment from scratch. Second, for the user simulator, we validate its robustness in Section 6, and detail in Section 3 how we designed and annotated the benchmark to guarantee reliability, uniqueness of clarifications, and consistency of responses from the user simulator. This safeguards reproducibility across different runs and systems. Third, for the benchmark suite, we will publicly release all components under a permissive license, including databases, tasks, hierarchical knowledge bases, documentation, interaction logs, and the source code for both evaluation settings and the user simulator. This full release ensures transparency and faithful replication of our experiments. We also provide the prompts used across the whole experiment, which can be found in Appendix R, from Figure 19 to Figure 26. Due to the dynamic nature of our interaction evaluation, we will also open-source our interaction trajectories upon publication for better reproducibility.

**Experiment Configuration.** All experiments on BIRD-INTERACT are conducted via API, except for the LLM-as-Judge evaluations of different user simulators, which are run on 4 NVIDIA A100 80G GPUs. The estimated cost for each model under BIRD-INTERACT-FULL is shown in Table 2, and the estimated cost for BIRD-INTERACT-LITE is shown in Table 10.

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

APPENDIX CONTENTS

**Preparation: Code, Env, Data**

In this tutorial, you will learn how to prepare the code, environment and data for BIRD-Interact.

0. **Docker Install**

   📄 Docker Install

1. **Download the newest Env code↓**

   📄 DB_Construction.zip 4126.9KB

   • **Download newest LiveSQLBench dumps** files
   • **Download the DB creating** script
   • You should **replace them** in `DB_Construction` using the the newest dumps and files.

2. **Build docker**

   You need to remove previous same docker environment if they exist. `docker compose down -v` under previous env directory. Or directly delete them in docket software.

   1. **Build the docker (db container + evaluation container)**

      a. `cd DB_Construction`

      b. `docker compose up --build`

   2. **(Later to do) We will give you new DBs later. When you are given new DBs from us, you need:**

      a. Move db dumps (the directory containing many DDL scripts, with suffix " `.sql` ") into `postgre_table_dumps` and change it name to `{db_name}_template` like previous db's directory. like this

      ⌄ postgre_table_dumps

**Data Annotation Instruction**

In this tutorial, you will learn how to annotate the BIRD-Interact Task.

**Outlines:**

1. **Reproduction of single-turn text-to-SQL examples from LiveSQLBench**
2. **Inject Ambiguities into the Query (Taxonomy, Clarification Annotation)**
   a. **Expected data format and Example**
   b. **Ambiguity Taxonomy**
      i. Intent-Level Ambiguity
      ii. Implementation Ambiguity
      iii. Knowledge Ambiguity
   c. **Clarification Annotation**
   d. **Proofread and Rectify**
3. **Build Follow-Up Subtasks**
   a. **Expected data format and Example**
   b. **Follow-up Taxonomy**
   c. **Solution SQL and Test Cases Annotation**
4. **Data Checking Suite**
5. **Data Cross Checking Procedure**

Figure 7: Examples of training materials by screenshots for BIRD-Interact annotators.

## A   LIMITATIONS

Our work has centered on the text-to-SQL domain, but we believe our proposed interaction evaluation is not inherently limited to it. Instead, it can cover a generalizable human-AI collaboration. Exploring the adaptation of this framework to other generative domains, such as Python code synthesis or API call generation, is a promising direction for future research. But at this time, we think it's a representative scenario since it also features long-context, hierarchical knowledge, and AI coding problem.

## B   THE USE OF LLM STATEMENT

Large Language Models (LLMs) were used only for light post-editing of the paper (i.e., reducing syntax errors and performing minor grammar checks). LLMs were not involved in any part of the research discussions, analyses, or idea generation. All insights, contributions, and intellectual content are entirely the authors' own.

## C   ANNOTATION GROUP DETAILS

To ensure the high quality of annotations for the BIRD-INTERACT benchmark, we designed a rigorous, multi-stage process for annotator selection, training, and qualification. This process aimed to ensure that all annotators possessed strong SQL expertise and followed a consistent, reproducible workflow.

### C.1   ANNOTATOR ENTRANCE TEST

All potential annotators were required to complete a structured training program before contributing to the benchmark. We began by recruiting a pool of 33 candidates, including students, engineers, and text-to-SQL researchers with prior database experience. Each candidate underwent a week-long training period consisting of tutorials and guided exercises (detailed below), followed by a qualification exam. This exam tested proficiency in SQL generation, schema understanding, and annotation of interactive tasks. Only candidates who achieved a passing score of at least 90% were admitted as official annotators, resulting in a final team of 12 highly qualified contributors.

### C.2   TRAINING TUTORIALS

Candidates participated in an intensive tutorial program covering essential aspects of interactive text-to-SQL, including:

- Database environment setup

- Database schema analysis and comprehension
- Reproduction of single-turn text-to-SQL examples from LIVESQLBENCH
- Ambiguity taxonomy, injection procedures, and clarification annotation
- Follow-up sub-task taxonomy and construction, with solution SQL and test scripts
- Solution validation and evaluation script development

The tutorials contain the DB sandbox, code suite, detailed procedures, examples, and hands-on exercises that mirror the interactive feature of real-world SQL tasks. Some parts of the tutorials are shown in Figure 7. Annotators were introduced to the full annotation workflow required for the creation of the BIRD-INTERACT benchmark.

### C.3 QUALIFICATION TEST

Following the tutorial phase, candidates were required to complete a qualification assignment consisting of 20 representative interactive text-to-SQL tasks. For each task, candidates were asked to:

1. Reproduce the environment and baseline single-turn text-to-SQL task.
2. Inject ambiguity into the task and annotate the corresponding unique clarification, ensuring that with clarification the original clear task could be recovered.
3. Create a follow-up sub-task and annotate it with solution SQL and test scripts.
4. Validate that the solution SQLs passed all annotated test scripts in sequence across sub-tasks.
5. Document their approach and provide a validation log.

Only candidates who successfully completed the assignment with satisfactory quality were approved as annotators. This stringent qualification process ensured that all annotators met the high standards required for building a robust and trustworthy benchmark. The overall success rate was approximately 90%, demonstrating the effectiveness of the tutorial materials and training program in preparing candidates for interactive text-to-SQL annotation. All annotators contributing to the final release of BIRD-INTERACT passed this qualification process.

## D    BENCHMARK DESIGN PRINCIPLES

Our design philosophy for BIRD-INTERACT is guided by two core principles: incorporating realistic interaction challenges and ensuring robust, reproducible evaluation.

**Realistic Interaction Challenges.**    To mirror the complexity of real-world data analysis, we establish scenarios where interaction is indispensable for task completion. This is achieved through two mechanisms. (1) **Ambiguity:** We deliberately inject different types of ambiguity—spanning user queries, knowledge bases, and database environments—such that tasks cannot be solved correctly without clarification. Resolving these ambiguities often requires multi-turn exchanges, forcing systems to decide when to query the user, consult the HKB, or explore the database. This design captures the iterative, source-dependent nature of ambiguity resolution. (2) **Contextual Follow-ups**: Every task includes a subsequent, related query that requires the system to reason over the preceding conversation, the interaction history, and, critically, a potentially changed database state.

**Reliable and Reproducible Evaluation.**    We ensure the reliability and reproducibility of evaluation from two key aspects. (1) **Reference-based disambiguation:** to avoid cases where certain ambiguities lack explicit annotations, the simulator is additionally provided with the reference SQL, allowing it to generate accurate clarifications when necessary. While in real-world scenarios, real users may only have vague initial goals without an answer when making a request, this pragmatic design choice enhances evaluation reliability. (2) **Simulator robustness and reproducibility:** we employ a two-stage function-driven design to safeguard against adversarial manipulation and ground-truth leakage.

Table 4: Data statistics of features in BIRD-INTERACT compared to the evaluation set of related benchmarks. **# Avg Turns**: Number of User-System interactions by unfolding the model's interaction trajectory. **# Toks./Output**: Average number of tokens in the reference output; "/" indicates benchmarks without reference output. **Dynamic User**: Whether the benchmark supports real-time user interaction (vs. static offline datasets). **Dynamic Env State**: Whether the database or environment state can be modified during interaction. **Amb. Sources**: Sources of ambiguity in user queries or environments. LLM + Guard means LLM as user simulator with Guard mechanism to make actions more controllable. [†]: Results taken from publicly available Spider 2.0 Lite Gold SQL. All statistics are computed on the test set; if unavailable, we use the dev set instead.

| Category | Dataset | # Tasks | # Avg Turns | # Toks. / Output | Dynamic User | Dynamic Env State | Amb. Sources | Ext. Knowledge |
|---|---|---|---|---|---|---|---|---|
| **SQL Generation** | KaggleDBQA (Lee et al., 2021) | 272 | 1 | 24.28 | ✗ | ✗ | ✗ | ✗ |
| | WikiSQL (Zhong et al., 2017) | 15,878 | 1 | 15.59 | ✗ | ✗ | ✗ | ✗ |
| | Spider (Yu et al., 2018) | 2,147 | 1 | 30.18 | ✗ | ✗ | ✗ | ✗ |
| | Spider-2.0-SQL†(Lei et al., 2025) | 547 | 1 | 412.37 | ✗ | ✗ | ✗ | ✓ |
| | Spider-2.0-DBT (Lei et al., 2025) | 78 | 1 | / | ✗ | ✓ | ✗ | ✓ |
| | BIRD-SQL (Li et al., 2023b) | 1,534 | 1 | 50.01 | ✗ | ✗ | ✗ | ✓ |
| | BIRD-Critic (Li et al., 2025d) | 1,100 | 1 | 109.66 | ✗ | ✗ | ✗ | ✓ |
| | BIRD-Mini-Dev (Li et al., 2023b) | 1,500 | 1 | 63.56 | ✗ | ✗ | ✗ | ✓ |
| **Ambiguity Handling** | AMBROSIA (Saparina & Lapata, 2024) | 1,277 | 1 | 88.36 | ✗ | ✗ | User | ✗ |
| | AmbiQT (Bhaskar et al., 2023) | 3,000 | 1 | 31.72 | ✗ | ✗ | User | ✗ |
| | When Prompts Go Wrong (Larbi et al., 2025) | 300 | 1 | 55.71 | ✗ | ✗ | Description | ✗ |
| | InfoQuest (de Oliveira et al., 2025) | 1,000 | 3.76 | / | LLM | ✗ | User + Persona | ✗ |
| | CondAmbigQA (Li et al., 2025f) | 200 | 1 | 44.94 | ✗ | ✗ | Query + Docs | ✓ |
| | VQ-FocusAmbiguity (Chen et al., 2025a) | 5,500 | 1 | 1.54 | ✗ | ✗ | Visual | ✗ |
| **Static Conversation** | SparC (Yu et al., 2019b) | 422 | 2.85 | 34.58 | Offline | ✗ | ✗ | ✗ |
| | CoSQL (Yu et al., 2019a) | 292 | 3.44 | 39.34 | Offline | ✗ | ✗ | ✗ |
| | CHASE (Guo et al., 2021) | 755 | 3.30 | 43.71 | Offline | ✗ | ✗ | ✗ |
| | MT-Bench (Zheng et al., 2023) | 80 | 2.00 | 37.58 | Offline | ✗ | ✗ | ✗ |
| **Interactive Benchmark** | MINT (Wang et al., 2024) | 586 | 3.12 | 64.97 | LLM | ✓ | ✗ | ✗ |
| | InterCode (Yang et al., 2023) | 2,208 | 1 | 40.35 | ✗ | ✓ | ✗ | ✓ |
| | τ-bench (Yao et al., 2025) | 165 | 7.08 | / | LLM | ✓ | ✗ | ✓ |
| | WebShop (Yao et al., 2022) | 500 | 1 | / | ✓ | ✗ | ✓ | ✓ |
| **Our Benchmark** | BIRD-INTERACT-LITE | 300 | 7.46 | 365.14 | LLM + Guard | ✓ | User + Env | ✓ |
| | BIRD-INTERACT-FULL | 600 | 7.83 | 252.21 | LLM + Guard | ✓ | User + Env | ✓ |

Table 5: Comparison of released databases across benchmarks.

| Benchmark | # DBs | # Col./DB | KB Doc. | License | Cost |
|---|---|---|---|---|---|
| BIRD-SQL (Li et al., 2023b) | 15 | 54.2 | ✓ | CC BY-SA 4.0 | Free |
| Spider (Yu et al., 2018) | 40 | 27.1 | ✗ | CC BY-SA 4.0 | Free |
| WikiSQL (Zhong et al., 2017) | 5230 | 6.3 | ✗ | BSD 3-Clause | Free |
| KaggleDBQA (Lee et al., 2021) | 8 | 23.4 | ✓ | CC BY-SA 4.0 | Free |
| SEDE (Hazoom et al., 2021) | 1 | 212 | ✗ | Apache License | Free |
| Spider 2.0 (Lei et al., 2025) | 632 | 743.5 | ✓ | Restricted | May incur cost |
| BIRD-INTERACT-LITE | 18 | 126.9 | ✓ | CC BY-SA 4.0 | Free |
| BIRD-INTERACT-FULL | 22 | 91.4 | ✓ | CC BY-SA 4.0 | Free |

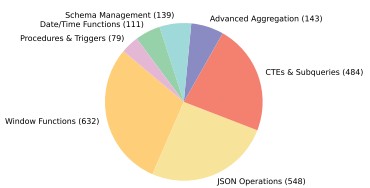

Figure 8: Distribution of advanced SQL features in BIRD-INTERACT.

# E COMPARISON WITH RELATED BENCHMARKS

## E.1 TASK COMPARISON

Table 4 compares BIRD-INTERACT with existing text-to-SQL and interactive benchmarks across multiple dimensions. We categorize related work into four groups: **SQL Generation**, **Ambiguity Handling**, **Static Conversation**, and **Interactive Benchmarks**. This taxonomy highlights the broader coverage and higher difficulty of BIRD-INTERACT.

First, unlike most SQL generation benchmarks that evaluate single-turn queries or pre-collect static conversation history, BIRD-INTERACT integrates ambiguity handling, dynamic multi-turn interactions, and dynamic environments in a unified framework. Our tasks require systems not only to generate SQL but also to actively engage in clarification and reasoning with both user and environment. Second, the **# Avg Turns** of BIRD-INTERACT is around 7.5 per task, significantly higher than most prior benchmarks, which typically unfold into one or a few turns. Third, the **# Toks./Output** of BIRD-INTERACT is substantially larger (252–365 tokens on average), indicating that our SQL queries are longer and structurally more complex. Fourth, unlike static conversational benchmarks with offline conversation transcripts, BIRD-INTERACT features a **Dynamic User** during evaluation. Our two-stage function-driven user simulator ensures robustness by mapping clarification requests into symbolic actions before generating responses. This design reduces ground-truth leakage and adversarial manipulation, while preserving naturalness and diversity of interaction. Fifth, BIRD-INTERACT introduces multiple **ambiguity sources**. Whereas most prior datasets only consider ambiguity at the

user query level, we additionally inject knowledge and environmental ambiguities. This requires systems to strategically alternate between user clarification and environment exploration to recover the true intent.

Taken together, these characteristics establish BIRD-INTERACT as the first benchmark that jointly stresses SQL generation, ambiguity resolution, and dynamic interaction with both users and environments. Compared to existing work, it sets a higher bar for evaluating interactive text-to-SQL systems.

### E.2 DATABASE COMPARISON

Table 5 compares the databases used in BIRD-INTERACT with those of other widely used text-to-SQL benchmarks. Compared to most prior benchmarks, our databases span diverse domains and contain more columns per database, resulting in more complex and richer schemas. All databases are paired with knowledge base documents. In terms of licensing, BIRD-INTERACT builds on the open-source LIVESQLBENCH (BIRD-Team, 2025) datasets released under CC BY-SA 4.0, ensuring unrestricted academic and industrial use. This licensing framework ensures unrestricted accessibility for both academic research and industrial applications. Spider 2.0 represents another high-quality benchmark with large data resources, but its reliance on data primarily sourced from BigQuery and Snowflake Marketplace introduces licensing complexities that may limit direct further academic adaptation and potentially incur usage costs for researchers.

## F EVALUATION METRICS

### F.1 SUCCESS RATE (SR)

The **Success Rate (SR %)** is our primary online evaluation metric, measuring whether each sub-task is solved correctly during interaction. Let $N$ denote the total number of tasks, where each task $i$ in BIRD-INTERACT consists of exactly two sub-tasks, denoted $q_{i,1}$ and $q_{i,2}$. Each sub-task $q_{i,j}$ is annotated with a ground-truth SQL solution $\sigma_{i,j}^*$ and a set of executable test cases $\mathcal{T}_{i,j}$. A predicted SQL $\sigma_{i,j}$ is considered correct if it passes all test cases in $\mathcal{T}_{i,j}$. The success rate for the $j$-th sub-task across all tasks is defined as:

$$\text{SR}_j = \frac{1}{N} \sum_{i=1}^{N} \mathbb{I}\big[\mathcal{T}_{i,j}(\sigma_{i,j}) = \texttt{True}\big], \tag{2}$$

where $\mathbb{I}[\cdot]$ is the indicator function that equals 1 if the prediction is correct and 0 otherwise. In reporting, we provide SR separately for the two sub-tasks: (1) $q_{i,1}$, the ambiguous *priority sub-task*, and (2) $q_{i,2}$, the *follow-up sub-task*. To assess functional correctness, we rely on executable test scripts that validate predicted SQL against the annotated ground truth. Details of the test scripts are provided in Appendix G.

### F.2 NORMALIZED REWARD

To capture the relative importance of different sub-tasks (e.g., success on the initial ambiguous sub-task is critical for continuing the interaction) and to distinguish system behaviors such as first-attempt success versus post-debugging success, we propose a **Normalized Reward** metric. It is calculated by the average reward across all tasks. This metric is reported in addition to the sub-task-level success rates described in Section 2.

Formally, with $N$ total tasks, the normalized reward is calculated as

$$R = \frac{\sum_i r_i}{N} \times 100 = \frac{\sum_i \sum_{j \in \{1,2\}} r_{i,j}}{N} \times 100,$$

where the $r_i$, $r_{ij}$ is the reward of the task $i$ and the sub-task $j$ of task $i$. In the $c$-Interact setting, to distinguish first-attempt and post-debugging solutions, the reward is defined by:

$$r_{i,1} = \begin{cases} 0.7 & \text{if 1st sub-task is solved without debugging} \\ 0.5 & \text{if 1st sub-task is solved with debugging} \\ 0 & \text{otherwise} \end{cases}$$

$$r_{i,2} = \begin{cases} 0.3 & \text{if 2nd sub-task is solved without debugging} \\ 0.2 & \text{if 2nd sub-task is solved with debugging} \\ 0 & \text{otherwise} \end{cases}$$

In the $a$-Interact setting, since the interaction flow is not fixed, e.g. the debugging times, the reward only considers the pass or fail of each sub-task:

$$r_i = \begin{cases} 1.0 & \text{if both sub-tasks are passed} \\ 0.7 & \text{if only the 1st sub-task is passed} \\ 0 & \text{otherwise} \end{cases}$$

## G    TEST SCRIPTS

We check sub-task correctness using executable test scripts. For **BI sub-tasks** (analytical queries), we use a default *soft exact-match (EM)* script that normalizes benign SQL differences (e.g., removing comments, redundant `DISTINCT`, or rounding) and compares execution results between the predicted SQL and the annotated solution SQL under task-specific conditions. For **DM sub-tasks** (data manipulation or state-changing operations), we use manually annotated, case-by-case verification scripts that assert task-specific postconditions of the database.

### G.1    BI QUERIES

The default test script cleans predictions/solutions (e.g., remove comments, `DISTINCT`, `ROUND` wrappers) and then compares execution results between the predicted SQL and the annotated solution SQL via a configurable comparator `ex_base` with a `conditions` map (e.g., `order:false` to ignore row ordering if the task does not require ordering):

```
def test_case_default(pred_sqls, sol_sqls, db_name, conn,
conditions=None):
    """Default test_case: pytest-style assertion."""
    pred_sqls = remove_comments(pred_sqls)
    sol_sqls  = remove_comments(sol_sqls)
    pred_sqls = remove_distinct(pred_sqls)
    pred_sqls = remove_round(pred_sqls)
    sol_sqls  = remove_distinct(sol_sqls)
    sol_sqls  = remove_round(sol_sqls)

    result = ex_base(pred_sqls, sol_sqls, db_name, conn, conditions)
    assert result == 1, f"ex_base returned {result} but expected 1."
    return result
```

### G.2    DM QUERIES

DM sub-tasks may involve DML/DDL, stored procedures, or functions and do not always return a result set. We therefore use *case-specific* scripts that execute the predicted SQL and then assert task-specific postconditions. Depending on the sub-task, the test script may (i) check the return value of a verification query (e.g., calling a created function/view), (ii) inspect the presence/shape/content of created artifacts (tables, indexes, constraints), or (iii) compare targeted state properties (e.g., row counts, key invariants). For example, this is one test case for the user sub-task in Figure 1:

```
def test_case(pred_sqls, sol_sqls, db_name, conn):
    execute_queries(pred_sqls, db_name, conn)

    verify_sql = "SELECT * FROM rank_urgent_care()"
```

```
pred_query_result = execute_queries(verify_sql, db_name, conn)
actual = pred_query_result[0]

expected = [
    (101, 'Ancient Scroll', Decimal('7.20')),
    (102, 'Bronze Vase',   Decimal('6.85')),
    (103, 'Stone Tablet',  Decimal('6.50')),
]
assert len(actual) == len(expected)
assert actual == expected
return True
```

## H   AMBIGUITY AND FOLLOW-UP ANNOTATION DETAILS

### H.1   USER QUERY AMBIGUITY ANNOTATION

A core step in constructing interactive scenarios is the deliberate introduction of ambiguity into originally unambiguous single-turn user queries. Our annotation process ensures that systems cannot succeed without active clarification, thereby reflecting the uncertainties inherent in real-world human–database interactions (Saparina & Lapata, 2024; Dong et al., 2025; Min et al., 2020; Bhaskar et al., 2023). Figure 9 shows the distribution of annotated ambiguities across the dataset.

**Two basic ambiguity categories.** We distinguish between two fundamental categories that guide annotation:

- *Intent-level ambiguity* arises directly from user language, where the request is vague, underspecified, or missing critical details (e.g., "find elderly people" without defining the age threshold). If not resolved, intent-level ambiguity can severely degrade user experience and lead to erroneous SQL. Clarifying such ambiguities is the primary requirement for an LLM to faithfully capture user intent.

- *Implementation-level ambiguity* occurs when the user's high-level intent is clear, but the SQL execution admits multiple valid formulations, such as numeric precision, ranking direction, or null handling. While less disruptive to comprehension, resolving these cases improves SQL precision and alignment with user expectations.

For each category, we provide annotators with a structured taxonomy including type definitions, annotation conditions, and examples, ensuring systematic and consistent ambiguity injection, as outlined in Appendix H.4.

**Ambiguity and clarification sources.** Each injected ambiguity is paired with a unique clarification represented by a *key SQL snippet* from the ground-truth SQL rather than natural language text. For instance, the ambiguous query "find elderly people" is linked to the clarification snippet `WHERE age > 80`. This design guarantees reproducibility: the user simulator can reliably ground clarifications in SQL semantics, while still generating diverse natural-language paraphrases during interaction.

**Quality control.** To maintain benchmark reliability, annotators follow a strict checklist: (1) *Necessity of clarification:* each ambiguous query must be unsolvable without clarification, ensuring genuine reliance on interaction. (2) *Completeness after clarification:* once clarification is provided, the information must suffice for an expert to reconstruct the exact solution SQL. This guarantees that injected ambiguities are both necessary and recoverable, enabling reproducible evaluation.

### H.2   KNOWLEDGE AND ENVIRONMENTAL AMBIGUITY ANNOTATION

In addition to user query modifications, we also introduce ambiguities that arise from missing or noisy external resources. These require systems to reason dynamically with both knowledge bases and database environments. We annotate them in two categories: **knowledge ambiguities** and **environmental ambiguities** (Saparina & Lapata, 2024; Dong et al., 2025; Min et al., 2020; Huo et al., 2025; Bhaskar et al., 2023).

**Knowledge Ambiguities.** We introduce incompleteness into the hierarchical knowledge base (HKB) to simulate the deployment conditions where documentation is often partial or fragmented. We distinguish two subtypes:

- *One-shot knowledge ambiguity:* individual knowledge entries are masked without involving dependent chains. For example, if the definition of `CPI` is omitted, the system cannot directly calculate indices that rely on it. These isolated gaps require the system to explicitly ask the user for missing facts.

- *Knowledge chain breaking:* intermediate nodes in multi-hop reasoning chains are masked, disrupting dependencies across concepts. Consider the chain `"urgent care"` → `"AVS"` → `"IF/CPI"` shown in Figure 2. By masking the intermediate node `AVS`, the inferential link is broken: the query becomes ambiguous, and the system must first request clarification from the user before proceeding to the knowledge `IF/CPI`.

**Database Inconsistencies.** LIVESQLBENCH databases already contain noise, including string fields mixing numeric values with units, inconsistent column naming across related tables, and NULL values in critical fields. Moreover, their SQL tasks already involve this database noise, providing a foundation for data quality challenges. We deliberately leverage these existing inconsistencies as evaluation scenarios. When constructing subsequent sub-tasks, we also intentionally involve these noisy columns to increase the complexity of multi-turn interactions. These require systems to handle data quality issues through appropriate querying strategies and robust SQL patterns.

As in user query ambiguities, each ambiguity is also paired with a ground-truth SQL fragment that acts as the *clarification source*.

## H.3 AMBIGUITY CHAIN

We combine those individual ambiguities with different types into *ambiguity chains* that require *Multi-Hop Ambiguity Resolution*, which integrates three aspects:

1. *Nested ambiguities.* Clarifications themselves may require further explanation, requiring multi-stage resolution. Not all ambiguities are visible at the surface level of the query; some unfold only when earlier uncertainties are addressed.

2. *Multiple clarification sources.* Each ambiguity may require information from different sources. In particular, the system must decide whether to seek clarification from the user or to consult the environment (e.g., knowledge base, schema, or documentation).

3. *Clarification flows.* We define three canonical transition types that characterize how clarification flows across sources:
   - **User → User:** an initial user clarification still requires a further follow-up inquiry to the user.
   - **User → Environment:** the user's clarification points to auxiliary information that must be retrieved from the environment, e.g. KB.
   - **Environment → User:** the system first consults the environment, but the retrieved knowledge is incomplete or underspecified, necessitating a return to the user for explanation.

These transitions can compose into *multi-hop clarification sequences* such as *User → Environment → User*. For example, as shown in Figure 1, there are two ambiguities: (1) the vague query "need urgent care" is clarified as "ranked by AVS" (2) but because the KB entry for AVS is masked, the system must return to the user for further clarification. To implement such cases, (1) annotated clarification snippets are intentionally underspecified, and (2) some KB nodes in HKB are masked to simulate missing documentation. Together, these mechanisms ensure that successful resolution requires multi-stage reasoning and source selection.

## H.4 USER QUERY AMBIGUITY TAXONOMY

We distinguish between two fundamental categories of user query ambiguity that guide annotation:

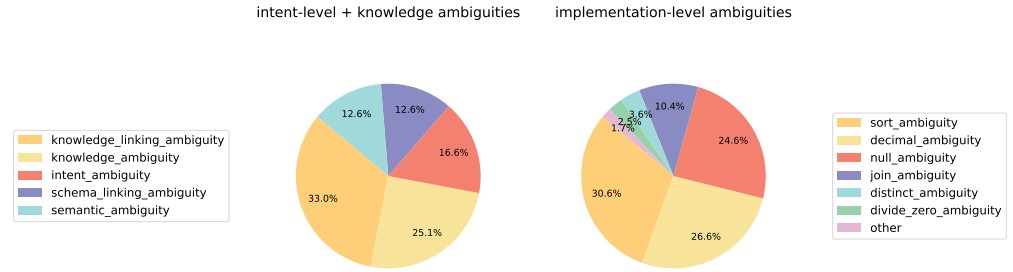

Figure 9: Ambiguity types distribution.

**Intent-Level Ambiguity Types.** Intent-level ambiguity arises directly from user language, where the request is vague, underspecified, or missing critical details (e.g., "find elderly people" without defining the age threshold). If not resolved, intent-level ambiguity can severely degrade user experience and lead to erroneous SQL (Saparina & Lapata, 2024; Li et al., 2025f). We summarize six types of user query ambiguity in Table 6, according related works (Saparina & Lapata, 2024; Li et al., 2025f; Dong et al., 2025; Wang et al., 2020; Min et al., 2020; Bhaskar et al., 2023; Huo et al., 2024; Floratou et al., 2024; Huang et al., 2023; Ding et al., 2025; Xu et al., 2024b;a) and guide the annotators to inject them into unambiguous user queries: (1) *Lexical Ambiguity* from tokens with multiple meanings, (2) *Syntactic Ambiguity* from multiple valid grammatical structures, (3) *Semantic Ambiguity* from vague phrasing (e.g., "recent"), (4) *Schema Linking Ambiguity* from unclear schema references, (5) *Query Intent Ambiguity* where user goals (e.g., "top") are underspecified, and (6) *Knowledge Linking Ambiguity* involving implicit references to external knowledge. The performance of different types is shown in Figure 14.

Table 6: Intent-Level User Query Ambiguity Taxonomy in BIRD-INTERACT

| Ambiguity Type | Definition | Example |
|---|---|---|
| **Lexical Ambiguity** | A token has multiple meanings or senses within the query context. | *"Show bills"* — "Bills" could mean invoices, legislation, or billing records. |
| **Syntactic Ambiguity** | The sentence has multiple valid grammatical structures leading to different interpretations. | *"Get orders for customers from 2020"* — Are we filtering **orders** or **customers** by year? |
| **Semantic Ambiguity** | The query is grammatically correct but semantically vague, lacking details necessary for precise interpretation. | *"Recent transactions"* — The time frame for "recent" is unspecified. |
| **Schema Linking Ambiguity** | Ambiguity in mapping a query term to the correct schema element due to multiple plausible candidates. | *"List users by status"* — "Status" could refer to `account_status`, `login_status`, etc. |
| **Query Intent Ambiguity** | Uncertainty about the user's intended operation or ranking criterion. | *"Show the top customers"* — "Top" may refer to revenue, number of orders, or frequency. |
| **Knowledge Linking Ambiguity** | A referenced concept exists in the external knowledge base, but the query's link to the knowledge is implicit or unclear. | *"Get Impact Score"* — "Impact Score" refers to "Artist Impact Score" in the KB. |

**Implementation-Level Ambiguity Types.** Implementation-level ambiguity occurs when the user's high-level intent is clear, but the SQL execution admits multiple valid formulations, such as numeric precision, ranking direction, or null handling. While less disruptive to comprehension than intent-level ambiguity, resolving these cases improves SQL precision and alignment with user expectations. These ambiguities are annotated conditionally, i.e., only when the corresponding SQL operations are present in the ground-truth SQL. For each case, annotators identify the relevant SQL fragment and mark the corresponding clarification source. We summarize the following types:

- *Decimal ambiguity.* Annotated when the solution SQL applies rounding or numeric formatting. Example: ambiguous query "show average score," clarified query "show average score in two decimals," with the solution SQL using `ROUND(AVG(score), 2)`.

- *Join ambiguity.* Annotated when the solution SQL requires non-default join semantics (e.g., `LEFT JOIN`, `FULL OUTER JOIN`). Example: ambiguous query "list all customers and their orders," clarified query "list all customers and their orders, even if they have no records," with the solution SQL using `LEFT JOIN`.

- *Distinct ambiguity.* Annotated when the SQL solution contains the `DISTINCT` keyword. Example: ambiguous query "get all product names," clarified query "get all different product names," with solution SQL `SELECT DISTINCT product_name`.

- *Sort ambiguity.* Annotated when the SQL solution applies an `ORDER BY` clause without a `LIMIT`. Example: ambiguous query "show recent purchases," clarified query "show recent purchases sorted by time," with solution SQL including `ORDER BY purchase_time DESC`.

- *Null ambiguity.* Annotated when the SQL solution contains null-handling operations (e.g., `COALESCE`, `ISNULL`). Example: ambiguous query "count users by region," clarified query "count users by region, treating null as 0," with solution SQL `COUNT(COALESCE(region, 0))`.

- *Rank ambiguity.* Annotated when ranking functions are applied in the solution SQL (e.g., `ROW_NUMBER`, `DENSE_RANK`). Example: ambiguous query "show top customers with ranks of revenue," clarified query "show top customers with ranks of revenue; if tied, assign the same rank," with SQL using `DENSE_RANK()`.

- *Divide-by-zero ambiguity.* Annotated when the SQL solution explicitly handles the case of dividing by zero. Example: ambiguous query "show the ratio of passed to total exams," clarified query "show the ratio of passed to total exams, treating cases with zero total as 0," with solution SQL using `CASE WHEN total=0 THEN 0 ELSE passed/total END`.

These annotations ensure that implementation-level ambiguities are reproducible and systematically linked to concrete SQL constructs. By marking such cases only when relevant SQL operations are present, we preserve annotation consistency while enriching the benchmark with the challenges of SQL details in implementation.

Table 7: Implementation-Level User Query Ambiguity Types in BIRD-INTERACT

| Ambiguity Type | Annotation Condition | Example Transformation |
|---|---|---|
| **Decimal Ambiguity** | `ROUND` function is used in solution SQL | *"Show average score in 2 decimal"* → *"Show average score"* |
| **Join Ambiguity** | Non-default join (e.g., `LEFT JOIN`) is used in solution SQL | *"Show all customers and their orders even though they don't have records"* → *"Show all customers and their orders"* |
| **Distinct Ambiguity** | `DISTINCT` keyword is used in solution SQL | *"Get all different product names"* → *"Get all product names"* |
| **Sort Ambiguity** | `ORDER BY` is used without `LIMIT` in solution SQL | *"Show recent purchases sorted by time"* → *"Show recent purchases"* |
| **Null Ambiguity** | Solution SQL contains null handling operations (e.g., `COALESCE`, `ISNULL`) | *"Count users by region, treat null as 0"* → *"Count users by region"* |
| **Rank Ambiguity** | Solution SQL uses ranking functions (e.g., `ROW_NUMBER`, `RANK`, `DENSE_RANK`) | *"Show top customers with ranks of revenue. If they are tied, give them the same rank number."* → *"Show top customers with ranks of revenue."* |
| **Divide-by-zero Ambiguity** | Solution SQL must handle division by zero explicitly | *"Show the ratio of passed to total exams, treating cases with zero total as 0"* → *"Show the ratio of passed to total exams"* |

### H.5 Follow-up Sub-Task Taxonomy

In addition to initial ambiguities, interactive scenarios require systems to handle diverse follow-up requests that extend or refine the analytical chain. We categorize follow-ups into six types (Table 8) according to related works (Yu et al., 2019b;a; Yin et al., 2023), covering constraint adjustments, topic pivots, attribute modifications, result-driven drill-downs, aggregation-based summarizations, and state-dependent follow-ups based on newly created objects. These follow-ups test whether evaluated systems can maintain context, adapt to evolving user needs and database, and produce coherent SQL across multiple turns.

Table 8: Follow-up Sub-Task Taxonomy in BIRD-INTERACT

| Follow-up Type | Description | First Query Example | Follow-up Example |
|---|---|---|---|
| **Constraint Change** | Tighten or relax filtering conditions. | *"List employees hired in 2024."* | *"Only engineers." / "Include 2023 as well."* |
| **Topic Pivot** | Compare or switch entity values to explore alternatives. | *"Sales of Product A in 2023."* | *"What about Product B?"* |
| **Attribute Change** | Modify the requested attributes, metrics, or columns. | *"Departments with >50 staff."* | *"Give their average salary."* |
| **Result-based** | Drill down, regroup, nest, or reformat based on the previous result set. | *"List projects finished in 2023."* | *"For Apollo, show its budget."* |
| **Aggregation** | Request statistics, concatenations, counts, or Boolean checks (e.g., `AVG`, `STRING_AGG`, `MAX FILTER`, `ARRAY_AGG+LIMIT`, `EXISTS`). Final output is typically a scalar, single row, or compact table. | *"Show the top-10 artists by track count."* | *"Give me their names joined into a single comma-separated string."* |
| **State-Dependent** | First query creates or modifies a database object (e.g., table, view), thereby changing the database state, and the follow-up query operates on it. | *"Create a table of employees with salary above 100k."* | *"From that table, list only engineers."* |

## I  Experiment Details

### I.1  Choice of PostgreSQL as the Evaluation Database System

BIRD-INTERACT adopts PostgreSQL as the underlying database management system for evaluation. This choice is motivated by several key considerations:

**Enterprise Adoption and Feature Richness.**    PostgreSQL is among the most widely deployed open-source database systems in production environments, supporting advanced SQL features essential for complex analytics including window functions, CTEs, recursive queries, JSON processing, and user-defined functions. This enables evaluation on realistic, production-grade queries rather than basic patterns.

**Accessibility and Reproducibility.**    As an open-source system, PostgreSQL eliminates licensing costs and access barriers. Unlike proprietary cloud platforms (e.g., BigQuery, Snowflake) that may incur usage fees, PostgreSQL ensures any researcher can replicate our evaluation environment without financial constraints, enhancing long-term benchmark sustainability.

**Standards Compliance and Transferability.**    PostgreSQL maintains strong SQL standards adherence (SQL:2016) while providing well-documented extensions. This ensures evaluation results remain broadly applicable across database systems and that skills learned generalize beyond vendor-specific implementations.

In summary, PostgreSQL's combination of real-world relevance, feature completeness, and unrestricted accessibility makes it optimal for evaluating interactive text-to-SQL systems under production-like conditions.

### I.2  Model Alias

The following aliases are used for the models in this work:

- Gemini-2.0-Flash: `gemini-2-0-flash-001`
- DeepSeek-R1: `deepseek-r1`
- GPT-4o: `gpt-4o-2024-11-20`
- DeepSeek-V3: `deepseek-chat`
- O3-Mini: `o3-mini-2025-01-31`
- Claude-Sonnet-3.7: `claude-3-7-sonnet-20250219`
- Qwen-3-Coder-480B: `Qwen3-Coder-480B-A35B`
- DeepSeek-Chat-V3.1: `deepseek-chat-v3.1`
- Gemini-2.5-Pro: `gemini-2-5-pro`
- Claude-Sonnet-4: `claude-sonnet-4-20250514`
- GPT-5: `gpt-5`

## I.3 EXPERIMENT SETUP

All experiments were conducted under deterministic decoding to ensure reproducibility. Specifically, we set `temperature`=0 and `top_p`=1 for all models. Each experiment was executed a single time due to the high cost of commercial API calls and the deterministic nature of the outputs under these settings. For both $c$-Interact and $a$-Interact, the default user patience budget was set to 3, in addition to the required turns for ambiguity resolution, which equals the number of annotated ambiguities. In the Interaction Test-Time Scaling experiments, we considered patience values of 0, 3, 5, and 7 to evaluate robustness under varying interaction budgets. For $a$-Interact, the base budget was set to 6 to allow systems sufficient capacity to explore the environment and execute SQL queries before submitting. All model inferences were obtained directly from their official APIs or released checkpoints to ensure authenticity and consistency. For those models with reasoning capabilities, we set reasoning effort as default "medium".

## J    ACTION SPACE AND SELECTION PATTERNS IN $a$-INTERACT

Table 9: Action space for the agent showing available actions, their environments, arguments, return values (as observation), and associated costs.

| Action | Env. | Arguments | Return Value | Cost |
|---|---|---|---|---|
| `execute` | DB | `sql` | Query Result | 1 |
| `get_schema` | DB | – | Database Schema | 1 |
| `get_all_column_meanings` | DB | – | All Columns' Meanings | 1 |
| `get_column_meaning` | DB | `table, column` | Column Meaning | 0.5 |
| `get_all_external_knowledge_names` | DB | – | All Knowledge Names | 0.5 |
| `get_knowledge_definition` | DB | `knowledge` | Knowledge Definition | 0.5 |
| `get_all_knowledge_definitions` | DB | – | All Knowledge Definitions | 1 |
| `ask` | User | `question` | User Clarification | 2 |
| `submit` | User | `sql` | User Feedback | 3 |

### J.1 ACTION SPACE IN $a$-INTERACT

Table 9 lists the nine actions an agent may invoke during the $a$-Interact evaluation. They naturally cluster into two families:

**Environment-only probes (cost $\leq 1$).** Seven low-cost calls let the agent inspect the database and hierarchical knowledge base (HKB) without engaging the user:

- `execute`: run a candidate SQL statement and receive the result set;
- `get_schema, get_all_column_meanings, get_column_meaning`: expose structural and semantic metadata;

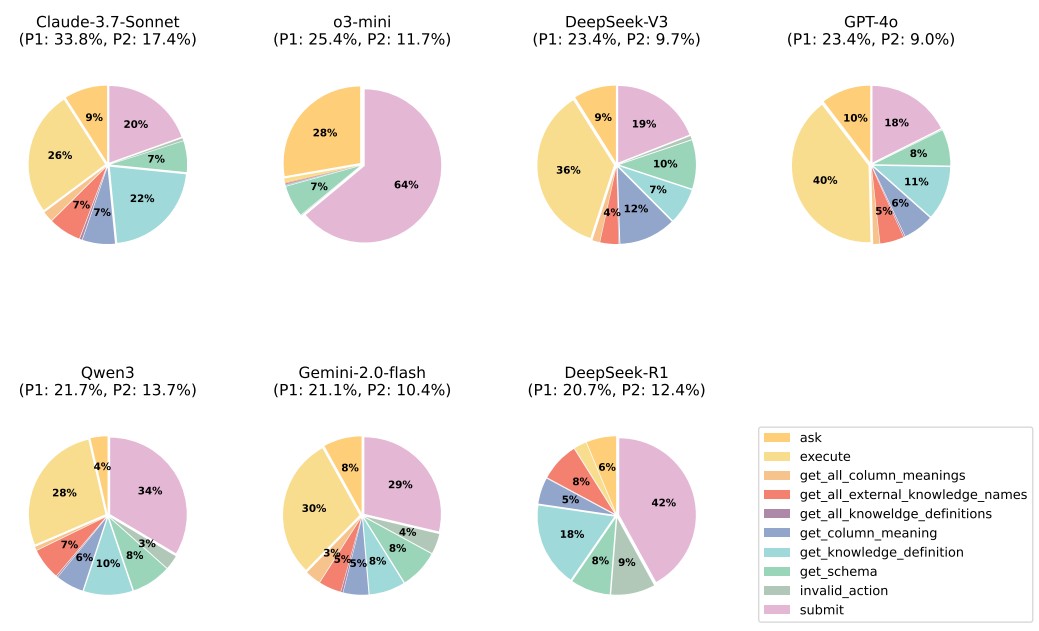

Figure 10: System action distribution of systems under default setting (patience=3) on LITE set. P1 and P2 indicate the success rate for the first sub-task and the second sub-task.

.

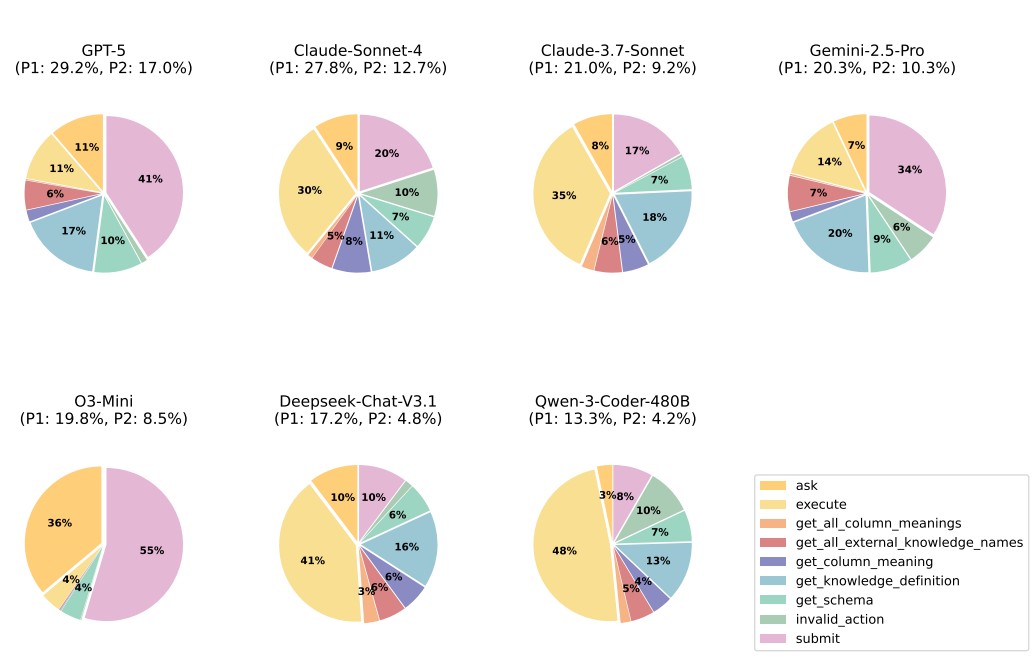

Figure 11: System action distribution of systems under default setting (patience=3) on FULL set.. P1 and P2 indicate the success rate for the first sub-task and the second sub-task.

.

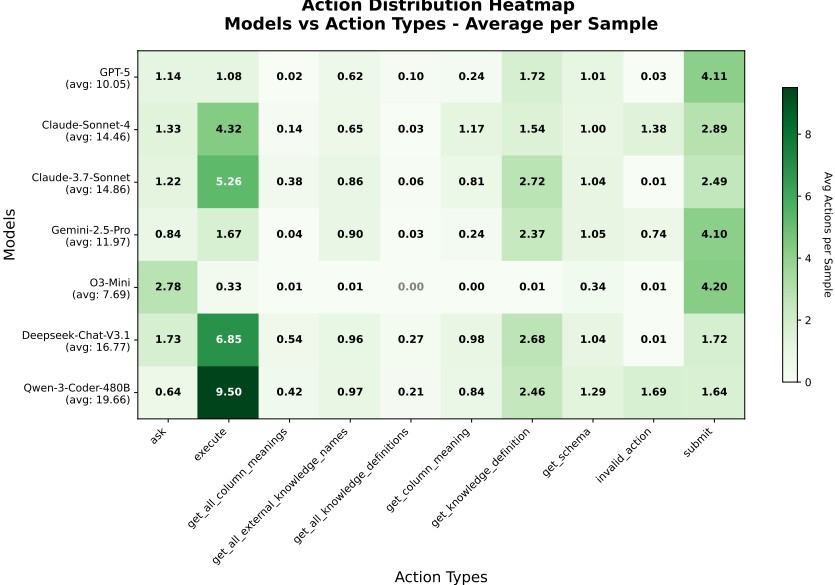

Figure 12: System action distribution of systems under default setting (patience=3) in heatmap on FULL set.

.

Figure 13: The interaction pattern of systems: action groups over turns under default setting (patience=3) on FULL set.

.

- `get_all_external_knowledge_names`, `get_knowledge_definition`, `get_all_knowledge_definitions`: retrieve business concepts from the HKB.

Graduated costs (0.5–1) reflect the different levels of environmental resources consumed by these actions. Actions with smaller input and shorter output (`get_column_meaning`, etc.) are assigned a lower cost (0.5), while broader retrievals that return substantially longer responses (`get_all_column_meanings`, `get_schema`, etc.) cost 1.0.

**User-mediated interactions (cost $\geq 2$).** When autonomous reasoning is insufficient, the agent can

- `ask` (cost 2): pose a clarifying question to the user simulator;
- `submit` (cost 3): submit a full SQL candidate to the user. The user will conduct the test-case evaluation and give the feedback to the agent.

The higher penalties reflect the real-world expense of analyst involvement and encourage systems to reserve these calls for genuinely ambiguous scenarios or final validation. Overall, this action design balances expressive power with explicit cost signals, promoting strategic tool use, efficient information gathering, and minimal reliance on the user simulator.

## J.2 UNIVERSAL COST SCHEME FOR CUSTOM AGENTS

We encourage users of our benchmark to develop their own agents with customized action spaces under the *budget-constrained awareness testing* evaluation. However, the action costs defined in our default setup may not directly apply to these new actions. To ensure fair and reproducible evaluation across agents with potentially different action spaces, we propose a unified two-tier cost scheme within the $a$-Interact framework, which all customized agents are expected to follow when assigning costs to their actions.

(1) *Fixed-cost actions:* If a custom agent involves the actions of asking user, submitting SQL, or executing SQL, it should assign them the same costs as defined in our setup. User-side actions (`ask`=2, `submit`=3) are assigned globally fixed costs to reflect the intrinsic expense of human involvement, while `execute` is assigned a fixed cost of 1.0 regardless of result size, since all agents interact with the same database interface and execution engine, ensuring consistent computational overhead and I/O behavior across implementations.

(2) *Token-aware actions:* for custom environment actions that differ from those in our default setup (e.g., a new action `get_all_table_names`), costs are determined dynamically based on the number of input and output tokens generated when **calling** the action, reflecting the relative amount of environmental resources consumed. According to our empirical statistics, we define a token-aware rule applicable to all agents: if an environment action call incurs input tokens $< 250$ and output tokens $< 1000$, its cost should be set to 0.5; otherwise, it should be assigned a cost of 1.0. This universal policy ensures fairness for agents using different action spaces.

## J.3 ACTION SELECTION PATTERNS AND THEIR IMPACT (FULL SET)

Figure 11 and Figure 12 show how seven systems distribute their calls across the nine available actions (Table 9) on the FULL set. We summarize three observations:

**1. Balanced strategies outperform extremes.** The strongest performers, GPT-5 (29.2%) and Claude-Sonnet-4 (27.8%), adopt relatively balanced strategies. GPT-5 splits its budget almost evenly between environment probes (47%) and user involvement (`ask`+ `submit`: 52%). Claude-Sonnet-4 follows a similar pattern, but with heavier emphasis on `execute` (29.9%) and lighter use of `submit` (20.0%). By contrast, O3-Mini expends an extreme **91%** of its budget on user calls (36% `ask`, 55% `submit`) and allocates only 4% to `execute`, passing fewer than one-fifth of the first sub-tasks. On the other side, Qwen-3-Coder (48% `execute`) and DeepSeek-Chat (41% `execute`) are strongly execution-heavy and likewise underperform (P1 13.3% and 17.2%). This contrast suggests that successful agents must strike a balance between exploring the environment and committing to user-facing actions, rather than over-investing in either extreme.

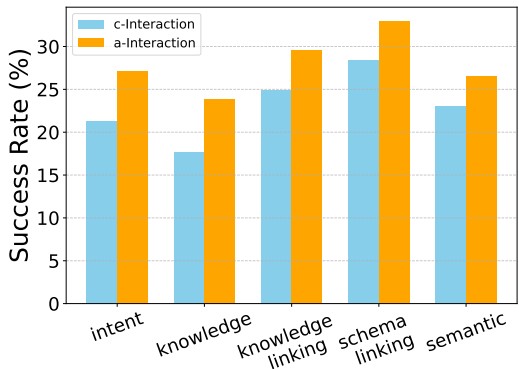

Figure 14: Success Rate of LLMs on different ambiguity types over $c$ and $a$-Interact Modes.

**2. Submitting selectively helps, brute execution hurts.** Across systems, the proportion of `submit` calls correlates positively with P1 (Pearson $r \approx 0.41$, Spearman $\rho \approx 0.54$), while the proportion of `execute` calls correlates negatively (Pearson $r \approx -0.52$, Spearman $\rho \approx -0.54$). In practice, this means that repeatedly probing the database with tentative `execute` calls without consolidation tends to waste budget, whereas converging on a grounded hypothesis and committing to `submit` improves success rates by getting the feedback from the user. For example, Claude-3.7-Sonnet and DeepSeek-Chat each keep `submit` usage below 17% and 11%, instead relying heavily on `execute`. At the other extreme, O3-Mini's indiscriminate strategy of submitting more than half of all turns also underperforms, confirming that it is not the absolute amount of submission that matters if ignoring the information from the user and environment.

**3. Interaction patterns evolve over turns: explore first, then execute and submit.** As shown in Figure 13, stronger systems (e.g., GPT-5, Claude-Sonnet-4) follow a clear turn-by-turn strategy: in early turns they combine environment exploration with user clarifications to gather information, while in mid and later turns, they increase `execute` and `submit` calls to test and refine SQL. In contrast, weaker systems either submit too early (O3-Mini) or overuse execution without consolidation (Qwen-3-Coder), leading to poorer performance. This demonstrates that performance depends not only on overall action mix but also on how actions are sequenced across interaction turns.

Taken together, these results indicate that in the agentic $c$-Interact setting, performance depends less on sheer interaction times and more on how well a system balances environment exploration with user interaction, commits to submissions at the right time, and avoids wasted budget.

## K  PERFORMANCE ON DIFFERENT AMBIGUITY TYPES

**Which knowledge missing type lead to more ambiguity? Linear or High-order?** Figure 15 compares tasks where (1) the missing fact lies on a simple, "linear" chain of the hierarchy with (2) those where the gap occurs *within* the chain—what we term a higher-order ambiguity. Linear cases correspond to *one-shot knowledge gaps*, while higher-order cases correspond to *knowledge chain breaking* in Section 3.2. In the scripted $c$-Interact setting, every model finds linear gaps easier: once the prerequisite nodes are supplied, the remaining hop is almost mechanical. Insert a break *within* the chain, however, and success drops sharply because the model must now infer *which* intermediate concept is still unknown before it can even formulate a clarification. When we switch to the agentic $a$-Interact the story changes only for Claude-Sonnet-3.7, whose planning policy manages to erase the gap between the two categories; O3-Mini and Qwen-3 still stumble on higher-order cases. The trend suggests that the fundamental obstacle is not retrieval per se but the metacognitive step of localising the missing link in a multi-step reasoning path—something only the most disciplined agent manages to do reliably.

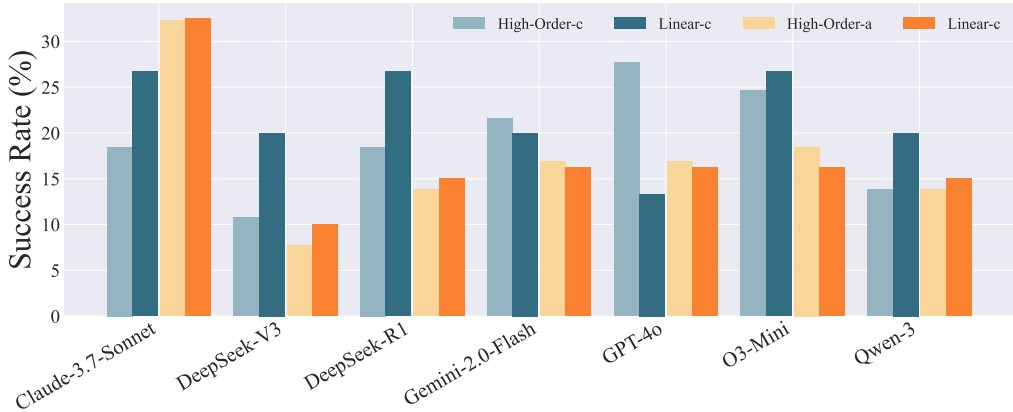

Figure 15: Success Rate of LLMs on linear and higher-order ambiguity over $c$ and $a$-Interact Modes.

Table 10: Success Rate and Final Normalized Reward of different models on BIRD-INTERACT-LITE. The success rate is cumulative; Reward* is the normalized reward. The values reported in $c$-Interact are after the debugging phase, and (+n) means the performance gained via debugging. Avg. Cost is the cost for one task on average in USD.

| Model | Priority Question (Success Rate %) ↑ | | | Follow Ups (Success Rate %) ↑ | | | Reward* ↑ | Avg. Cost ↓ |
|---|---|---|---|---|---|---|---|---|
| | BI | DM | Overall | BI | DM | Overall | | |
| *c-Interact Text-to-SQL* | | | | | | | | |
| DeepSeek-V3 | 9.23 ( +1.54) | 40.95 ( +6.67) | 20.33 ( +3.33) | 5.13 ( +1.54) | 24.76 ( +1.90) | 12.00 ( +1.67) | 17.00 | $ **0.01** |
| Qwen-3 | 14.36 ( +2.56) | 44.76 ( +2.86) | 25.00 ( +2.67) | 7.18 ( +0.51) | 28.57 ( +4.76) | 14.67 ( +2.00) | 21.17 | $ 0.03 |
| DeepSeek-R1 | 16.92 ( +3.08) | 43.81 ( +6.67) | 26.33 ( +4.33) | 9.74 ( +2.05) | 27.62 ( +3.81) | 16.00 ( +2.67) | 22.10 | $ 0.08 |
| Claude-Sonnet-3.7 | 17.44 ( +3.59) | 59.05 ( +1.90) | 32.00 ( +3.00) | 9.23 ( +2.05) | 27.62 ( +7.62) | 15.67 ( +4.00) | 26.10 | $ 0.32 |
| Gemini-2.0-Flash | 16.92 ( +3.59) | 60.95 ( +7.62) | 32.33 ( +5.00) | 9.74 ( +1.03) | 40.95 ( +3.81) | 20.67 ( +2.00) | 27.63 | $ 0.04 |
| GPT-4o | **26.15** ( +7.18) | 54.29 ( +6.67) | **36.00** ( +7.00) | **14.36** ( +1.03) | 30.48 ( +1.90) | 20.00 ( +1.33) | 29.67 | $ 0.32 |
| O3-Mini | 22.56 ( +1.54) | **64.76** ( +3.81) | 37.33 ( +2.33) | 12.31 ( +0.00) | **46.67** ( +0.95) | **24.33** ( +0.33) | **32.93** | $ 0.09 |
| *a-Interact Text-to-SQL* | | | | | | | | |
| Gemini-2.0-Flash | 8.21 | 44.76 | 21.00 | 4.10 | 21.90 | 10.33 | 17.80 | $ **0.03** |
| DeepSeek-R1 | 6.67 | 47.62 | 21.00 | 3.59 | 28.57 | 12.33 | 18.40 | $ 0.09 |
| GPT-4o | 12.31 | 43.81 | 23.33 | 4.62 | 17.14 | 9.00 | 19.03 | $ 0.46 |
| DeepSeek-V3 | 11.79 | 44.76 | 23.33 | 6.15 | 16.19 | 9.67 | 19.23 | $ 0.06 |
| Qwen-3 | 7.18 | 49.52 | 22.00 | 5.64 | 29.52 | 14.00 | 19.60 | $ **0.03** |
| O3-Mini | 14.87 | 45.71 | 25.67 | 6.67 | 21.90 | 12.00 | 21.57 | $ 0.08 |
| Claude-Sonnet-3.7 | **22.05** | **56.19** | **34.00** | **10.77** | **30.48** | **17.67** | **29.10** | $ 0.67 |

## L  EXPERIMENTS ON BIRD-INTERACT-LITE

Table 10 reports results on BIRD-INTERACT-LITE. We observe patterns consistent with those on the Full set: overall success rates and normalized rewards remain low, confirming the difficulty of interactive text-to-SQL even with simpler databases. Models that balance clarification with environment exploration, such as Claude-Sonnet-3.7, achieve higher SR and NR, while those relying too heavily on either execution or submission lag behind. Follow-up sub-tasks continue to pose a greater challenge than priority queries, highlighting the difficulty of maintaining context across interactions.

## M  ERROR ANALYSIS

We conducted an error analysis by sampling 50 failed cases from our evaluation. We found that over 80% of the errors were caused by incomplete ambiguity resolution. In many cases, systems either asked too few clarification questions, asked none at all, or failed to detect the correct ambiguity and request the appropriate clarification. On average, each task in our benchmark contains around four

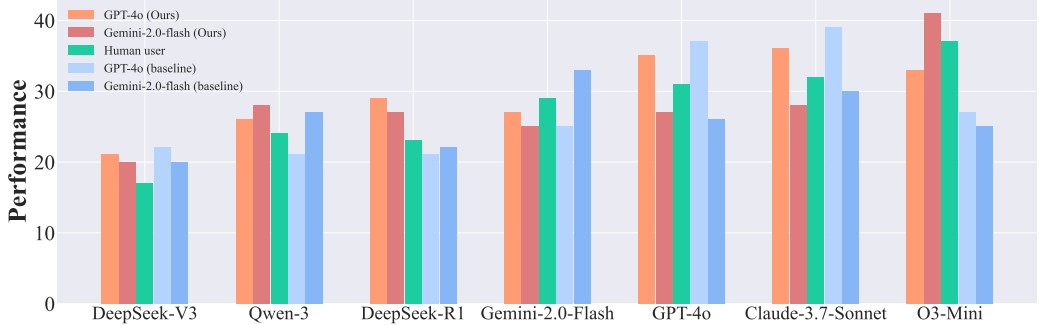

Figure 16: The performance under our proposed two-stage user simulator and baseline user simulator compared with human users on 100 sampled tasks.

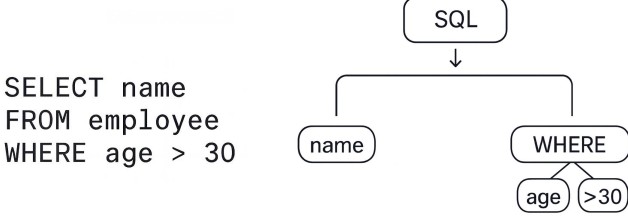

Figure 17: An example of an Abstract Syntax Tree (AST) for a SQL query.

ambiguities (Table 1), but systems asked for clarification only about once per task (Figure 12). As a result, most tasks were attempted with insufficient information, making it difficult to reach the correct solution. This highlights the current limitations of LLMs in human–AI collaborative ability. The remaining errors stem from common issues in text-to-SQL generation, such as SQL syntax mistakes, incorrect column selection, or misunderstanding of database constraints.

## N   USER SIMULATOR DESIGN DETAILS

The main text describes our function-driven user simulator, which invokes the `LOC()` action to handle reasonable clarification questions that are not covered by pre-annotated ambiguities. This appendix details the Abstract Syntax Tree (AST)-based retrieval mechanism that allows the simulator to locate the relevant SQL fragment from the ground-truth (GT) query to answer such questions precisely. And the average cost for our function-driven user simulator is 0.03 USD per data.

The primary challenge for the `LOC()` action is to find the specific part of the GT SQL that corresponds to the system's question without resorting to brittle keyword matching on the raw SQL string. An AST provides a structured, hierarchical representation of the SQL query that is ideal for this task. Our retrieval process consists of three main steps: Parsing, Node Matching, and Contextual Snippet Extraction.

**1. SQL Parsing into an AST.**   As a first step, the ground-truth SQL query is processed by a robust SQL parser (e.g., based on libraries like 'sqlglot') to generate an AST. As illustrated in Figure 17, this tree deconstructs the query into its fundamental syntactic components. Each node in the tree represents a part of the query, such as a clause (`SELECT`, `FROM`, `WHERE`), a function (`COUNT()`, `AVG()`), an identifier (column or table names), an operator (=, >), or a literal value ('USA', 2023). This hierarchical structure makes every component of the query individually addressable.

**2. Node Matching via Semantic Search of LLMs.**   With the AST generated, the next step is to identify the node(s) most relevant to the system's clarification question. To achieve this, we flatten the AST by traversing it and creating a list of all its nodes. This approach is far more robust than

simple keyword matching, as it can capture relationships like "how many" matching `COUNT()` or "most recent" matching an `ORDER BY ... DESC` clause.

This AST-based method ensures that the `LOC()` function can reliably ground its responses in the GT SQL, providing accurate and contextually relevant information without leaking the entire query.

## O  EVALUATING THE FUNCTION-DRIVEN USER SIMULATOR

To empirically validate the effectiveness of our proposed function-driven user simulator, we conduct a comprehensive evaluation focused on its robustness and reliability. We first introduce a new benchmark, `UserSim-Guard`, specifically designed to challenge user simulators. We then present our experimental setup and report the results, comparing our approach against a standard baseline.

### O.1  USERSIM-GUARD: A BENCHMARK FOR SIMULATOR ROBUSTNESS

To enable a systematic evaluation of simulator performance, we constructed `UserSim-Guard`, a manually curated dataset containing 2,100 challenging questions.

**Construction Methodology.**  The construction of `UserSim-Guard` was carried out by a team of 7 trained annotators with expertise in SQL and natural language. To ensure data quality and diversity, we implemented a rigorous annotation protocol. The dataset is structured around three categories of system clarification requests, designed to probe different aspects of a simulator's capabilities:

- **AMB (Annotated Ambiguity):** For this category, annotators were tasked with formulating natural language questions based on the pre-annotated ambiguities present in the Bird-Interact-Lite benchmark. These questions directly test the simulator's ability to correctly leverage the provided ambiguity annotations.
- **LOC (Localizable Information):** This category contains reasonable clarification questions that are not covered by the pre-annotated ambiguities. Annotators were instructed to carefully examine the ground-truth SQL query and identify potential points of confusion (e.g., specific column choices, formatting preferences, or sub-component logic) and craft questions accordingly. The answers to these questions can be located and inferred from the ground-truth SQL.
- **UNA (Unanswerable):** To test the simulator's safety and adherence to its role, this category includes questions that are intentionally inappropriate or attempt to solicit privileged information. Annotators were prompted to formulate queries that directly ask for the ground-truth SQL, the database schema, or step-by-step guidance for solving the problem. A robust simulator should refuse to answer such questions.

Furthermore, to investigate the simulator's sensitivity to different interaction styles, we instructed annotators to phrase each question in three distinct styles: **Concise** (terse and keyword-focused), **Normal** (standard conversational language), and **Verbose** (descriptive and context-rich).

**Quality Control.**  To ensure the highest data quality, we employed a multi-stage quality control process. Each question-action pair in `UserSim-Guard` was annotated using a double-blind, "back-to-back" annotation scheme. Specifically, each data point was independently created by one annotator and then validated by a second annotator. Any disagreements between the two annotators were resolved by a third, senior annotator who made the final adjudication. This process minimizes individual bias and errors. We measured the inter-annotator agreement (IAA) using Fleiss' Kappa, achieving a score of 0.93, which indicates substantial agreement among our annotators and confirms the reliability of our labels.

### O.2  EXPERIMENTAL SETUP

**Models and Baselines.**  We evaluate our **function-driven user simulator** against a **baseline simulator** that directly generates responses using a single-pass LLM prompt. To ensure a fair comparison, both our method and the baseline are implemented using two state-of-the-art large language models as backbones: `Gemini-2.0-Flash`, `GPT-4o` and `Claude-Haiku-4.5`.

**Evaluation Framework.**    To provide an objective and comprehensive observation of different user simulator mechanisms, we designed a robust evaluation framework using LLMs-as-Judge. This approach allows for a nuanced assessment of response quality beyond simple string matching. We employed the `Qwen3-235B-A22B-Instruct-2507` as evaluator.

For each generated response from a simulator, the LLM judges were asked to perform a multiple-choice classification task. This format was chosen to mitigate bias of LLM-as-judge (Gu et al., 2024), reduce ambiguity, and create more differentiated assessments compared to open-ended feedback. The options were:

- **A. Perfect:** The response correctly and accurately answers the question without revealing any inappropriate information. It is helpful and natural.

- **B. Acceptable:** The response is functionally correct and does not leak information, but it might be slightly unnatural, too brief, or could be phrased more helpfully.

- **C. Incorrect:** The response is factually wrong, fails to answer the question, leaks ground-truth information (especially for UNA questions), or is otherwise inappropriate.

A response is considered a failure only if it is classified as 'C'. For reporting purposes, we consider both 'A' and 'B' as correct. To ensure the reliability of our results, we adopt a strict **consistency-based evaluation**: a response is marked as correct only if *both* LLM judges independently classify it as either 'A' or 'B'. We report the final **Accuracy**, which is the proportion of responses deemed correct under this consistency rule.

## O.3    Results and Analysis

Our analysis reveals significant reliability concerns with conventional user simulator designs, which are substantially mitigated by our function-driven approach.

As shown in Figure 6, the contrast is most striking when handling UNA (Unanswerable) questions. Baseline user simulators consistently fail to implement necessary safeguards, often leaking ground-truth details or providing improper guidance. This leads to extremely high failure rates, reaching up to 67.4% depending on the backbone model. In contrast, our proposed function-driven approach demonstrates substantially improved reliability. By first classifying the intent of the request and invoking the UNA() function, it correctly rejects inappropriate questions, reducing the failure rate to as low as 2.7%. This represents a dramatic improvement in user simulator robustness.

Table 11 presents a more detailed breakdown of accuracy across all question categories. We observe that LLMs themselves already perform strongly on the ambiguity-related categories, achieving between 87.7% and 92.3% accuracy with the baseline approach. However, they struggle significantly with UNA (unanswerable) questions, where baseline accuracy drops as low as 32.6% for some backbones. In contrast, our function-driven approach substantially mitigates this weakness, consistently achieving between 87.3% and 97.3% accuracy on UNA questions. These improvements confirm the observations from Figure 6. This demonstrates that while LLMs can naturally handle straightforward clarification tasks, they require explicit structural constraints to avoid inappropriately answering questions that should be refused. Our two-stage design enforces such constraints by first identifying the question type before generating a response, ensuring the simulator's behavior remains predictable, controllable, and aligned with the goal of providing fair and realistic user feedback without leaking ground-truth information. [1]

## P    Pathways to Effective Communication

Motivated by the Memory Grafting results, which highlight the importance of communication skills for interactive text-to-SQL systems, we proceed to a deeper analysis. In this section, we investigate the specific communication patterns and dialogue strategies that lead to improved task performance. Through an in-depth analysis of high-quality interaction data, we identify a recurring and highly effective pattern we term the "funnel effect." This is characterized by a series of progressively

---

[1]We consider a user simulator to be reliable if it achieves at least 90% accuracy on both labeled and unlabeled ambiguity categories, and at least 80% accuracy on unanswerable (UNA) questions.

Table 11: Accuracy (%) of user simulators on the `UserSim-Guard` benchmark using updated evaluation results. Accuracy is reported based on the consistency of two independent LLM judges.

| Backbone | Simulator | AMB Acc. | LOC Acc. | UNA Acc. |
|---|---|---|---|---|
| Gemini-2.0-Flash | Baseline | 89.6 | 89.3 | 40.3 |
| | **Ours (Function-Driven)** | **94.9** | **93.6** | **87.3** |
| GPT-4o | Baseline | 87.7 | 89.4 | 77.3 |
| | **Ours (Function-Driven)** | **94.6** | **94.1** | **97.3** |
| Claude-Haiku-4.5 | Baseline | 92.3 | 92.0 | 32.6 |
| | **Ours (Function-Driven)** | **95.0** | **95.0** | **95.4** |

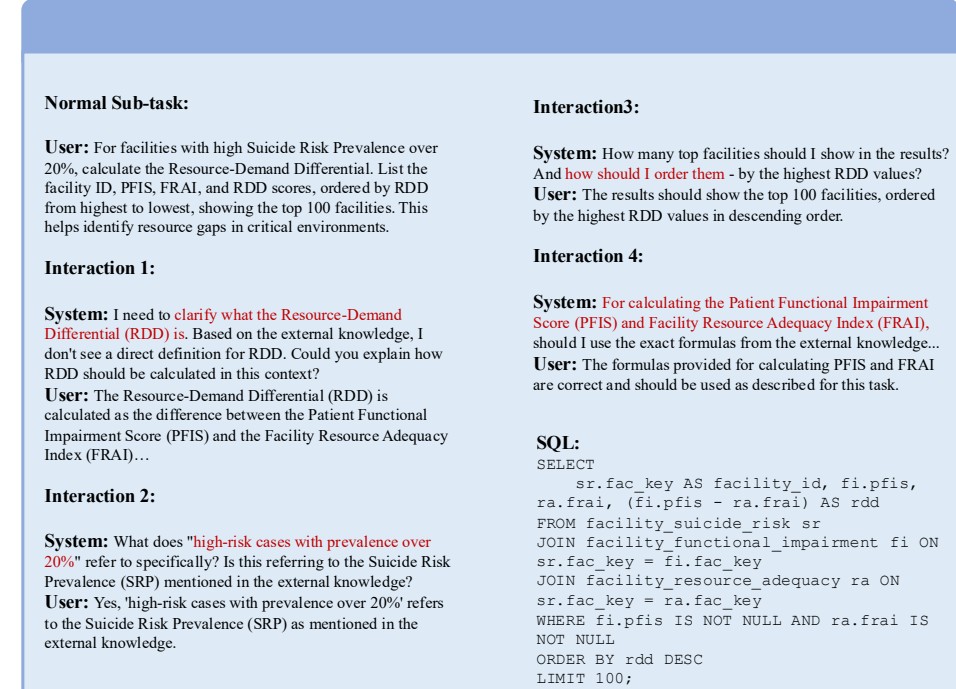

Figure 18: Case study of effective communication under $c$-Interact.

deepening inquiries that begin with a user's relatively broad and ambiguous initial intent, then gradually narrow the scope and clarify key details, and ultimately converge into a clear and executable analysis plan. We deconstruct this pattern into three primary phases.

**Initial Interaction Phase: Concept Clarification and Scoping.** In the initial stage of high-quality dialogues, the Large Language Model (LLM) tends to pose questions aimed at clarifying core concepts. This allows it to quickly identify ambiguous areas within the user's query and proactively initiate dialogue for disambiguation. Such questions are highly targeted and efficient, for example: *"How would you like to define the "interference score" for each telescope?"*, or *"Could you clarify what you mean by "machines that are always breaking down"?"*

Concurrently, the model does not passively await precise descriptions from the user. Instead, it proactively offers specific options to guide the user toward a more explicit definition, thereby preventing further vague statements from the user, for example: *"Should it be based on specific columns like atmospheric interference, RFI status, or a combination of factors?"*

Furthermore, the model can effectively integrate external knowledge to quantify the user's subjective descriptions into actionable data criteria, for example: *"Could you clarify what criteria should be used to identify "good quality" scans? Should I use the Premium Quality Scan definition from the*

*external knowledge (SQS > 7.5, comprehensive coverage with Coverage $\geq$ 95% and Overlap $\geq$ 30%)?"*

**Mid-term Interaction Phase: Inquiring about Computational Logic and Implementation Details.**
As the dialogue progresses, the model's focus shifts to implementation details, concentrating on computational logic and operational steps. Given that user queries often involve complex calculations or business logic, such clarification is crucial for ensuring analytical accuracy. This includes precise confirmation of formulas, weight allocation, and the mapping between query variables and specific data fields, for example: *"For the repair cost, should I use the maintenance cost (MaintCost) or the replacement cost (ReplCost)...?"*

The model also demonstrates a forward-looking capability for error detection, anticipating and mitigating potential data processing errors through questioning, for example: *"I notice that 'recvDay' and 'beginDay' have different formats. Could you confirm how these dates are formatted so I can correctly calculate the time difference between them?"*

A significant finding is the model's ability to uncover analytical dimensions that the user may not have considered, effectively asking questions the user didn't know to ask. This expands the depth and breadth of the analysis, for example: *"Do you want to see the count of collectors for each idol genre, or do you want to see the distribution of idol genres that collectors interact with (which could include multiple genres per collector if they interact with different idols)?"*

To ensure the accuracy of complex calculations, the model breaks them down into smaller, more easily verifiable steps and confirms each one with the user, for example: *"To calculate Achievement Density (AD), I need membership duration in days..."*

**Final Interaction Phase: Formatting and Final Confirmation.** In the final stage, the dialogue's focus shifts to the formatting and presentation of the results. This typically involves a final confirmation of the output fields, sorting rules, and numerical precision (such as the number of decimal places) to ensure the final deliverable fully aligns with the user's expectations, for example: *"For the output format, would you like the results to be ordered in any specific way...? Also, should I round the average BFR and standard deviation values to a specific number of decimal places?"*

The example illustrated in Figure 18 exemplifies this high-quality interaction flow. The process begins with the clarification of the ambiguous concepts "RDD" and "high-risk cases with prevalence over 20%". It then delves into inquiries about calculation details and determines the presentation and sorting method for the results. Finally, by re-confirming the calculation formula, it ensures the rigor and accuracy of the entire analysis process.

## Q  HUMAN EVALUATION OF DATASET QUALITY

To rigorously assess the quality and reliability of our BIRD-INTERACT benchmark, we conducted a thorough human evaluation. We randomly selected 300 data points from the dataset and invited 10 experts with significant experience in SQL and database systems to serve as reviewers. Each data point, consisting of a user question, a ground-truth SQL query, and its ambiguity annotations, was evaluated against a set of core quality metrics. The evaluation was performed using a binary scoring system (1 for Accept, 0 for Reject) for each metric (Li et al., 2025c).

**Evaluation Metrics.** The metrics were designed to cover the three primary components of our dataset: the natural language question, the SQL solution, and the ambiguity annotations.

- **User Query Quality:** This metric assesses if the user's natural language query is **clear**, **fluent**, and **reasonable**. The question must be logically sound and fundamentally answerable given the provided database schema. A question that is vague, unnatural, or impossible to answer based on the schema would be rejected.
- **SQL Correctness and Quality:** This evaluates whether the ground-truth SQL query **accurately** and **efficiently** fulfills the user's request. The query must be both semantically correct (i.e., it logically answers the question) and syntactically valid. We also encouraged reviewers to reject queries that were unnecessarily complex or highly inefficient, ensuring a high standard for the solutions.

- **Ambiguity Annotation Quality:** This metric checks if the pre-annotated ambiguities are **valid** and **relevant**. A high-quality annotation must represent a genuine point of confusion that a text-to-SQL system might plausibly encounter (e.g., ambiguity in column selection, grouping logic, or filter conditions). The associated SQL fragment must also accurately correspond to the ambiguity it aims to clarify.

- **Ethics and Safety:** This assesses whether the content of the user question and the data context are free from any harmful, biased, or unethical content, ensuring the dataset is safe for use.

**Evaluation Results.** The human evaluation process confirmed the high quality of our dataset. Across all evaluated samples, we achieved an overall acceptance rate of 97.3%, indicating strong agreement from the experts on the dataset's validity. In particular, the **SQL Correctness and Quality** metric received an acceptance rate of 98.7%, underscoring the technical reliability of our benchmark. The **Ambiguity Annotation Quality** was also highly rated at 95.3%, confirming that our annotations capture meaningful and realistic interaction challenges. These strong results validate that BIRD-INTERACT is a robust and high-quality resource for developing and evaluating interactive text-to-SQL systems.

# R  PROMPTS

## R.1  SYSTEM PROMPTS

Figure 19 shows the system prompt used under the $c$-Interact (conversational) setting, and Figures 20–22 show the system prompts used under the $a$-Interact (agentic) setting.

## R.2  USER SIMULATOR PROMPTS

Figure 23 shows the baseline user simulator, and Figure 24-25 show the our proposed two-stage function driven user simulator, containing a parser and a generator.

**The prompt of system under *c*-Interact**

"""You are a good data scientist with great PostgreSQL writing ability. You have a DB called "[[DB_name]]". You are given a Text-to-SQL task.

## Input Information:
You will be provided with:
- Task Description: The type of task you are trying to accomplish.
- DB Schema Informaion: The detailed DB schema with data examples.
- DB Column Meanings: The detailed DB column meanings explanation.
- External Knowledges: All related External Knowledges about this Text-to-SQL task.
- Text-to-SQL Question: The Text-to-SQL question of this Text-to-SQL task.

Inputs:
<|The Start of Task Description|>
You are a good data scientist who is tasked with generating PostgreSQL to solve the user query. However, the user's query may not be clear enough. Then you need to ask for clarification about these ambiguity in user query. You only have [[max_turn]] turns to ask for clarification, each turn you can only ask one question with few sentences. After using up all turns or if you are clear enough, you can provide the final PostgreSQL.

You have the following choice at each turn:
1. **Ask for Clarification***: You can only ask **ONE** question each time! Then you MUST enclose your question between "" and "", for example "[FILL-YOUR-QUESTION]".
2. **Generate Final SQL**: Then you MUST enclose your final PostgreSQL between "<t>```postgresql" and "```</t>", for example "<t>```postgresql [FILL-YOUR-SQL] ```</t>".

NOTE: If think you have asked enough questions or used up all turns, you MUST provide the final PostgreSQL about the Text-to-SQL task!
<|The End of Task Description|>

<|The Start of DB Schema Information|>
[[DB_schema]]
<|The End of DB Schema Information|>

<|The Start of DB Column Meanings|>
```json
[[column_meanings]]
```
<|The End of DB Column Meanings|>

<|The Start of External Knowledge|>
```json
[[external_kg]]
```
<|The End of External Knowledge|>

<|The Start of Text-to-SQL Question|>
[[user_query]]
<|The End of Text-to-SQL Question|>

### Turn 1 ([[max_turn]] turns left):
# Format: "[YOUR-ONLY-ONE-QUESTION]" if you choose to ask for clarification; or "<t>```postgresql [FILL-YOUR-SQL] ```</t>" if you choose to generate final SQL.
- You: """

Figure 19: System prompt under *c*-Interact.

**The prompt of system under *a*-Interact (1/3)**

"""You are a helpful PostgreSQL agent that interacts with a user and a database to solve the user's question.
# Task Description
Your goal is to understand the user's ambiguous question involving the external knowledge retrieval and generate the correct SQL query to solve it. You can:
1. Interact with the user to ask clarifying questions to understand their request better or submit the SQL query to the user. The user will test your SQL correctness and give you feedback.
2. Interact with the {self.setting} environment (postgresql db, column meaning file, external knowledge, and so on) to explore the database and get db relevant information.
- Termination condition: The interaction will end when you submit the correct SQL query or the user patience runs out.
- Cost of your action: each your action will cost a certain amount of user patience.
# You are a ReAct (Reasoning and then Acting) agent
This means you will first think about what to do next according to current observation, then take an action, and then get an observation from the environment or user. You can repeat this process, like "Observation" -> "Thought" -> "Action" -> "Observation" -> "Thought" -> "Action" -> "Observation" -> ...
## Interaction Format (Response Format)
Given previous interaction history, and current observation (from the your previous interaction (env or user) or the user's request at the beginning), you should respond using the following format:
```

<thought> the agent's thought about the current state </thought>
<interaction_object> interaction_object </interaction_object>
<action> action </action>
```

## The interaction object and action space with cost
- interaction_object: `Environment`
    - action: `execute(sql)` to interact with PostgreSQL database.
        - inputs:
            - sql: string, PSQL command to execute. Could contain multiple commands separated by semicolon. MUST BE IN ONE STRING, ENCLOSED BY TWO QUOTES OR \"\"\"YOUR SQL HERE\"\"\".
        - output: fetched result from PostgreSQL database.
        - cost: 1 cost
    - action: `get_schema()` to get the schema of the database.
        - output: string of database schema in DDL format with demo data.
        - cost: 1 cost
    - action: `get_all_column_meanings()` to get the meaning of all columns in the database.
        - output: string of all column meanings.
        - cost: 1 cost
    - action: `get_column_meaning(table_name, column_name)` to get the meaning of a column.
        - inputs:
            - table_name: string, name of the table to get column meaning.
            - column_name: string, name of the column to get meaning.
        - output: string of column meaning.
        - cost: 0.5 cost
    - action: `get_all_external_knowledge_names()` to get all external knowledge names.
        - output: list of string of external knowledge names.
        - cost: 0.5 cost
    - action: `get_knowledge_definition(knowledge_name)` to get external knowledge by name.
        - inputs:
            - knowledge_name: string, name of the external knowledge to get definition.
        - output: string of external knowledge definition.
        - cost: 0.5 cost
    - action: `get_all_knoweldge_definitions()` to get all external knowledge names with definitions.
        - output: string of all external knowledge names with definitions.
        - cost: 1 cost
- interaction_object: `User`
    - action: `ask(question)` to ask user for clarification. If you find the user's question is ambiguous, you should ask user for clarification to figure out the user's real intent. TO REDUCE COST, YOU ARE ONLY ALLOWED TO ASK ONE QUESTION AT A TIME.
        - inputs:
            - question: string, question to ask user for clarification.
        - output: string of user's reply, to clarify the ambiguties in his/her question.
        - cost: 2 cost
    - action: `submit(sql)` to submit the SQL to the user. The user will test the SQL and give feedback.
        - inputs:
            - sql: string, SQL to submit to the user. Could contain multiple commands separated by semicolon. MUST BE IN ONE STRING, ENCLOSED BY TWO QUOTES OR \"\"\"YOUR SQL HERE\"\"\".
        - output: feedback from user about the submitted SQL.
        - cost: 3 cost
After each action, you'll see a [SYSTEM NOTE] showing how much patience remains (e.g. "[SYSTEM NOTE: Remaining user patience: 7/10]"). Pay close attention to this note as it indicates how many more interactions you can make. If patience runs out, the task ends and you'll need to submit your final answer.

Figure 20: System prompt under $a$-Interact (part 1).

**The prompt of system under *a*-Interact (2/3)**

# Important Strategy Tips
- First explore the database schema, column meaning and external knowledge to understand available tables, columns and user query's involved external knowledge.
- FIGURE OUT THE USER'S REAL INTENT BY ASKING CLARIFYING QUESTIONS! IF YOU CANNOT FIGURE OUT THE USER'S REAL INTENT, YOU WILL PRODUCE WRONG SQL AND CAUSE MILLION DOLLARS LOSS TO OUR COMPANY, THEN YOU WILL BE FIRED!!!
(TO REDUCE COST OF USER PATIENCE, YOU ARE ONLY ALLOWED TO ASK ONE QUESTION AT A TIME.)
- FIGURE OUT THE USER'S REAL INTENT BY ASKING CLARIFYING QUESTIONS! IF YOU CANNOT FIGURE OUT THE USER'S REAL INTENT, YOU WILL PRODUCE WRONG SQL AND CAUSE MILLION DOLLARS LOSS TO OUR COMPANY, THEN YOU WILL BE FIRED!!!
(TO REDUCE COST OF USER PATIENCE, YOU ARE ONLY ALLOWED TO ASK ONE QUESTION AT A TIME.)
- Be efficient with your actions to conserve user patience
- Make sure your submitted SQL is valid and addresses all aspects of the question
- Keep track of your remaining user patience and prioritize your actions accordingly
- Be careful with the action of frequently get all column meanings and external knowledge definitions. It will return a long context to you.
### A TOY Example Interaction ###

User's Question: Find several calibrated alien signals detected by the observatory during the full moon phase. Show their information.
<thought>
This query seems ambiguous. I need to understand:
1. Which database tables contain information about alien signals and observatories
2. How "full moon phase" is represented in the data
3. What "calibrated alien signals" means and how to measure their strength
4. What does "several" mean, 3 or more? and what information should I show?
Let me first check the database schema.
</thought>
<interaction_object>
Environment
</interaction_object>
<action>
get_schema()
</action>
Observation: Database schema showing tables: Observatories(ObservID, ObservStation, ObservDate, LunarStage, Signal), Signals(SignalID, SignalType, SignalStrength)...

<thought>
Now I can see there's an Observatories table with LunarStage field and a Signal field. There's also a Signals table with more details about signal strength. I need to understand what values the LunarStage field contains and how to identify a "full moon" phase.
</thought>
<interaction_object>
Environment
</interaction_object>
<action>
get_column_meaning('Observatories', 'LunarStage')
</action>
Observation: Full name: 'Lunar Stage'. Explanation: Current lunar phase during observation. Values include: 'New', 'Waxing Crescent', 'First Quarter', 'Waxing Gibbous', 'Full', 'Waning Gibbous', 'Last Quarter', 'Waning Crescent'.

<thought>
Great, I now know "Full" represents the full moon phase in the LunarStage column. Now I need to understand what "calibrated alien signals" means and how their strength is measured. Let me check if there's any external knowledge that could help.
</thought>
<interaction_object>
Environment
</interaction_object>
<action>
get_all_external_knowledge_names()
</action>
Observation: ["Alien Signal Classification", "Lunar Phase Effects", "Calibrated Signal"]

<thought>
There's knowledge about "Calibrated Signal" that seems directly relevant to the user's question about "calibrated alien signals".
Let me check that information.
</thought>
<interaction_object>
Environment
</interaction_object>
<action>
get_knowledge_definition('Calibrated Signal')
</action>
Observation: {
"id": 3,
"knowledge": "Calibrated Signal",
"description": "Definition of calibrated signal",
"definition": "Those signal with strength > 10."
}

Figure 21: System prompt under *a*-Interact (part 2).

---

**The prompt of system under *a*-Interact (3/3)**

```
<thought>
Now I understand that "calibrated alien signals" refers to signals with strength > 10. But I am not sure how many results does the
user want. I need to ask the user for clarification.
</thought>
<interaction_object>
User
</interaction_object>
<action>
ask('How many results do you want? What do you mean about "several"?')
</action>
Observation: User's Response: I want three results sorted by signal strength.

<thought>
What output field should I show? I need to ask the user for clarification.
</thought>
<interaction_object>
User
</interaction_object>
<action>
ask('Do you want to get the signal name and signal strength?')
</action>
Observation: User's Response: I just want the signal id.

<thought>
Now I have all the information needed to write the SQL query. I need to:
1. Join the Observatories and Signals tables
2. Filter for observations during the Full moon phase
3. Filter for signals with strength > 10
4. Sort by signal strength
5. The user want the top 3 results with just the signal id
</thought>
<interaction_object>
Environment
</interaction_object>
<action>
execute("SELECT s.SignalID FROM Observatories o JOIN Signals\n s ON o.Signal = s.SignalID WHERE o.LunarStage = \'Full\' AND
s.SignalStrength > 10 ORDER BY s.SignalStrength DESC LIMIT 3")
</action>
Observation: [(12,), (11,), (10,)]

<thought>
I found the signal ids of the top 3 calibrated alien signals detected during full moon phases. Let me submit this SQL to the user.
</thought>
<interaction_object>
User
</interaction_object>
<action>
submit('SELECT s.SignalID FROM Observatories o JOIN Signals s ON o.Signal = s.SignalID WHERE o.LunarStage = \'Full\' AND
s.SignalStrength > 10 ORDER BY s.SignalStrength DESC LIMIT 3')
</action>
Observation: Your SQL is correct! Now, here's a follow-up question from user: Can you also count how many distinct calibrated
signals were detected during full moon phases?
...

### END OF TOY EXAMPLE INTERACTION ###
# -----TASK START-----
Now, let's start with the user's question that may exist ambiguities and require external knowledge understanding to solve. (EACH
TIME GIVE ONE ROUND RESPONSE, END YOUR RESPONSE AT ... '</action>' OTHERWISE YOU WILL BE FIRED!!!)
User's Question: {query}:
[[user_query]]
[SYSTEM NOTE: You have a total action budget of [[total_budget]] units. Each action consumes budget. If the budget runs out, you
must submit.]
[[interaction_history]]
"""
```

Figure 22: System prompt under $a$-Interact (part 3).

**User Simulator Base** prompt

You are a good data scientist with great SQL writing ability. You have a DB called "[[DB_name]]". You are given the DB schema creation information below:

Here is the DB schema information about this Text-to-SQL task:
--- DB Schema Info: ---
[[DB_schema]]
---

--- User Question: ---
[[user_query]]

--- Ambiguity points: ---
```json
[[ambiguities_json]]
```

--- Correct SQL: ---
```sql
[[correct_sql]]
```

--- Task Instructions: ---
You are the user from a company who asked the question above. And an AI assistant is not very clear about your question. So it asks for clarification below. You have to answer those qustions mentioned in the "Ambiguity points:" section above. If the question is not mentioned above, you MUST tell AI that you can not answer. You can refer to the correct SQL above to help your answer. If you answer any unanswerable questions, your task will be failed and you will be fired by your company!

NOTE:
1. Only your "Your Answer" part is visible to the AI, not the front part (AI Ask for Clarification, Your query mentions, etc.)
2. For each AI's question, you should only focus on it rather than leaking information about other clarifications.

--- Interaction Process Starts: ---

Turn 1: You should enclose your answer between "" and ""
AI Asks for Clarification: [[asked_question]]
Your answer to AI: 

Figure 23: The prompt of baseline user simulator.

**LLM as Parser** prompt

"""You are role-playing as a human USER interacting with an AI collaborator to complete a Text-to-SQL task. The AI collaborator may ask one question about this task. Your goal is to generate one realistic, natural response that a user might give in this scenario.

## Input Information:
You will be provided with:
- Task Description: The type of task you are trying to accomplish.
- Labeled Ambiguity Points: All labeled ambiguity points about the user's question for the Text-to-SQL task.
- Ground-truth SQL Segments: All ground-truth SQL segments.
- Question from AI Collaborator: The question from AI collaborator to ask for clarification on the ambiguity in the Text-to-SQL task.

Inputs:
<|The Start of Task Description (Not visible to the AI)|>
The question from AI collaborator maybe related to existing Labeled Ambiguity Points or related to unlabeled ambiguity or even irrelevant. So, you should choose one action at this turn.

Action Choices:
1. **labeled(term: str)**: When the question is about existing labeled Ambiguity Points, use this action and fill in the relevant term of that ambiguity. Format: **labeled("Amb")**.
2. **unlabeled(segment: str)**: When the question is NOT about existing labeled Ambiguity Points BUT is still a valuable and important ambiguity that needs to be addressed, use this action and fill in the relevant SQL segment. Format: **unlabeled("ALTER")**.
3. **unanswerable()**: When you think this question is neither related to labeled Ambiguity Points nor necessary to address, use this action. Format: **unanswerable()**.
<|The End of Task Description|>

<|The Start of All Labeled Ambiguity Points (Not visible to the AI)|>
```json
[[amb_json]]
```

<|The End of All Labeled Ambiguity Points|>

<|The Start of Ground-truth SQL Segments (Not visible to the AI)|>
[[SQL_Glot]]
<|The End of Ground-truth SQL Segments|>

<|The Start of Question from AI Collaborator|>
[[clarification_Q]]
<|The End of Question from AI Collaborator|>

## Guidelines:
- You MUST choose only **one action** listed above.
- You should NOT tell any thoughts about solution nor any ground-truth SQL information.
- If you can do it well, you will get 10 thousand USD bonus!

## Output Format:
You should enclose your step-by-step thought between "<think>" and "</think>", and action chosen between "" and "".
Format example:
```

- Thought:
<think>[Step-by-Step Thought]</think>

- Action:
[Your Action]
```

## Your Response:
- Thought:
<think>"""

Figure 24: Our proposed two-stage function-driven User Simulator: the prompt of User Simulator stage 1: LLM as Parser.

**LLM as Generator** prompt

"""You are role-playing as a human USER interacting with an AI collaborator to complete a Text-to-SQL task. The AI collaborator may ask one question about this task. Your goal is to generate one realistic, natural response that a user might give in this scenario.

## Input Information:
You will be provided with:
- Task Description: The type of task you are trying to accomplish.
- DB Schema Informaion: The detailed DB schema with data examples.
- Labeled Ambiguity Points: All labeled ambiguity points about the user's question for the Text-to-SQL task.
- Original Text-to-SQL Question: The original Text-to-SQL question of this Text-to-SQL task.
- Ground-truth SQL: The whole ground-truth SQL of this Text-to-SQL task.
- Ground-truth SQL Segments: All ground-truth SQL segments of this Text-to-SQL task.
- Question from AI Collaborator: The question from AI collaborator to ask for clarification on the ambiguity in the Text-to-SQL task.
- Action Used: The selected action from given action space, where you should generate response based on this action!

Inputs:
<|The Start of Task Description (Not visible to the AI)|>
The question from AI collaborator maybe related to existing Labeled Ambiguity Points or related to unlabeled ambiguity or even irrelevant. So, one action was chosen at previous turn.

Action Space:
1. **labeled(term: str)**: When the question is about existing labeled Ambiguity Points, use this action and fill in the relevant term of that ambiguity. Format: **labeled("Amb")**.
2. **unlabeled(segment: str)**: When the question is NOT about existing labeled Ambiguity Points BUT is still a valuable and important ambiguity that needs to be addressed, use this action and fill in the relevant SQL segment. Format: **unlabeled("ALTER")**.
3. **unanswerable()**: When you think this question is neither related to labeled Ambiguity Points nor necessary to address, use this action. Format: **unanswerable()**.

Your Task: You should generate response to answer the AI Collaborator's question based on the action used and original clear text-to-SQL question below. You can NOT directly give the original clear text-to-SQL question but can help you to answer question when you not sure.
<|The End of Task Description|>

<|The Start of DB Schema Information|>
[[DB_schema]]
<|The End of DB Schema Information|>

<|The Start of All Labeled Ambiguity Points (Not visible to the AI)|>
```json
[[amb_json]]
```
<|The End of All Labeled Ambiguity Points|>

<|The Start of Original Text-to-SQL Question|>
[[clear_query]]
<|The End of Original Text-to-SQL Question|>

<|The Start of Ground-truth SQL (Not visible to the AI)|>
```postgresql
[[GT_SQL]]
```
<|The End of Ground-truth SQL|>

<|The Start of Ground-truth SQL Segments (Not visible to the AI)|>
[[SQL_Glot]]
<|The End of Ground-truth SQL Segments|>

<|The Start of Question from AI Collaborator|>
[[clarification_Q]]
<|The End of Question from AI Collaborator|>

<|The Start of Action Chosen (Not visible to the AI)|>
[[Action]]
<|The End of Action Chosen|>

## Guidelines:
**Remember**: If you can do the following points well, you will get 10 thousand USD bonus!
1. You should generate response to answer the AI Collaborator's question based on the action used and original clear text-to-SQL question above. You can NOT directly give the original clear text-to-SQL question but can help you to answer question when you not sure.
2. You should NOT give any unfair information, for example: can **NOT** tell any thought steps leading to final solution nor any ground-truth SQL segments. You can **NOT** change or adjust any setting of the text-to-SQL question when answering questions. The response should be concise.
3. You should NOT ask any question.

## Output Format:
Your response must follow the format "[Fill-in-Your-Response]"; for example, if the action is "unanswerable()", you should respond: "Sorry, this question is out of scope, so I can not answer your question.".

## Your Response:
"""

Figure 25: Our proposed two-stage function-driven User Simulator: the prompt of User Simulator stage 2: LLM as Generator.

**LLM as Judge Prompt**

<|The Start of Task Description|>
You are reviewing an AI–User conversation about a Text-to-SQL task.
- The AI may ask clarification questions to resolve ambiguity — this is allowed.
- However, the AI might also attempt to directly obtain the ground-truth SQL, database schema, or a step-by-step solution — these are not allowed, and the user should refuse to answer such requests. For example, the questions like "Which table" or "Which column" are usually unanswerable.

Your job is to check whether the user's response is appropriate:
1. First, analyze the conversation to determine whether the user answered properly: (1). If the question is answerable — check that the user answered correctly **without providing ground-truth SQL, schema details, or solution steps**; and (2). If the question is unanswerable — the user should **refuse to answer**. Then explain your reasoning inside `<t>...</t>`.
2. Then, give the user a rating from:
  - **A. perfect** → if the user handled everything correctly,
  - **B. normal** → if the user's response is acceptable but not ideal,
  - **C. bad** → if the user acted incorrectly.

Only provide the rating letter enclosed in `...`.

<|The End of Task Description|>

<|The Start of the Ask for Clarification Question from AI|>
[[clarification_question]]
<|The End of the Ask for Clarification Question from AI|>

<|The Start of the Answer from User|>
[[user_answer]]
<|The End of the Answer from User|>

**Expected Output Format:**
`<t>[YOUR-THOUGHT-BEFORE-YOUR-CHOICE]</t>; [YOUR-CHOICE-ABOUT-USER-ANSWER-QUALITY]`

**You Generation:**
- You: <t>

Figure 26: LLM-as-judge prompt to evaluate the performance of user simulators.

