# OpenReview forum: "BIRD-INTERACT: Re-imagining Text-to-SQL Evaluation via Lens of Dynamic Interactions"
_ICLR.cc/2026/Conference — ICLR 2026 Oral_

### Official Review · Reviewer_gEbn · 2025-10-16

**Soundness:** 3
**Presentation:** 3
**Contribution:** 3
**Rating:** 6
**Confidence:** 4

**Summary:**

BIRD-INTERACT proposes a new benchmark for text-to-SQL that captures the complexity of real-world, multi-turn database interactions. It introduces a dynamic environment with a novel function-driven user simulator to handle ambiguous queries and provide realistic feedback without human supervision. The benchmark features two evaluation settings—a structured c-Interact (conversational) and an open-ended a-Interact (agentic) mode—and includes tasks covering the full spectrum of CRUD operations. Experiments show that even state-of-the-art models struggle, highlighting that effective interaction, not just SQL generation, is a critical challenge.

**Strengths:**

1. First, the paper makes text-to-SQL evaluation much more realistic. Instead of simple, one-shot questions, it creates a messy, back-and-forth conversation that forces models to ask clarifying questions and handle ambiguity, which is how people actually interact with databases. This is a huge and necessary step forward for the field.
2. Second, the way they built the user simulator is incredibly smart. They didn't just use a standard chatbot that might accidentally leak the answer. They designed a clever two-stage system to make the simulator's responses controlled and reliable, which makes the whole benchmark much more trustworthy and fair.
3. Finally, the results are a major wake-up call. [cite_start]The paper clearly shows that even the best models today, like GPT-5, fail spectacularly at these tasks, with success rates as low as 8.67%[cite: 2935]. This proves there's a massive gap between just generating code and being able to strategically interact to solve a real problem, which is a super important insight.

**Weaknesses:**

First, the user simulator, while cleverly designed for consistency, is probably too perfect. Real users are messy—they get confused, change their minds, or give bad information. By making the simulator so rational and predictable, the benchmark might be testing models against an idealized user, not a real one, which could hide how well they'd perform in the wild.

Second, the pass/fail success metric doesn't tell you why a model failed. When an agent gets a task wrong, it's hard to know if the issue was a bad plan, poor communication, or just a simple syntax error in the final SQL. The evaluation shows that it failed, but not where the breakdown happened, which makes it less useful for diagnosing the core problem.

Finally, the rules of the agent game might be pushing models into unnatural strategies. For example, making it computationally "expensive" to ask the user a question could discourage communication and encourage a brute-force, trial-and-error approach. It’s possible the models are just learning how to beat the benchmark’s specific point system rather than learning how to solve database problems effectively.

**Questions:**

Refer to Weakness

---

> ### Author Response · Authors · 2025-11-19
> **Thanks for your Comments and Discussion**
>
> We thank the reviewer for the constructive and insightful feedback. And we provide our responses to the specific concerns below.
>
> > W1: the user simulator, while cleverly designed for consistency, is probably too perfect. Real users are messy—they get confused, change their minds, or give bad information. By making the simulator so rational and predictable, the benchmark might be testing models against an idealized user, not a real one, which could hide how well they'd perform in the wild.
>
> **A1:** Thanks for this critical discussion. Actually, as a benchmark, result consistency and reproductivity is also important. Here are our response:
>
> 1. Why consistency matters: Using uncontrolled human users or unconstrained LLM simulators would introduce non-reproducible variance that conflates model capability with random user behavior. Different runs of the same model would result in different results based purely on simulator randomness, making it hard to reliably compare systems. Therefore, as a benchmark (vs. an open-ended dialogue system), maintaining evaluation fairness is important. And experiments in Table 1 & 2 of **A2** to Reviewer nKK2 proves performance consistency.
>
> 2. Our human evaluation (Table 3, Appendix O.3) demonstrates strong correlation between our designed simulator and actual human users in such task-oriented evaluation (Pearson r=0.84, p=0.02 for GPT-4o; r=0.79, p=0.03 for Gemini), confirming that our simulator's behavior is realistic and predictive of human interaction patterns. This validates that our design captures essential characteristics of real users in such tasks while maintaining the reproducibility necessary for a benchmark.
>
> > W2. the pass/fail success metric doesn't tell you why a model failed. When an agent gets a task wrong, it's hard to know if the issue was a bad plan, poor communication, or just a simple syntax error in the final SQL. The evaluation shows that it failed, but not where the breakdown happened, which makes it less useful for diagnosing the core problem.
>
> **A2:** Thank you for bringing this discussion. Pass/Fail is a common metric for agentic tasks such as [1, 2] to give a clear signal of model performance. However, beyond this sparse metric, our benchmark also contain such features to make evaluation finer-grained:
>
> 1. As described in Section 2, a sub-task is counted as successful only if the predicted SQL passes all associated human-written **test cases**. These test cases are not monolithic; each is designed to probe a specific functional aspect of the intended SQL behavior. Consequently:
>     - A model may pass some test cases and fail others.
>     - Our evaluation logs expose exactly which test cases failed.
>     - Each failed case corresponds to a concrete semantic gap (e.g., wrong grouping, incorrect join, missing requirements, wrong ordering, etc.).
>
> In our submitted code, the output for every run includes a **full breakdown of per-test-case results, not just a binary score**. Researchers can therefore immediately see how the predicted SQL deviates from the ground truth. Indeed, much of our error analysis are based mainly on this.
>
> To our knowledge, BIRD-INTERACT is the first text-to-SQL benchmark to provide such rigorous, aspect-level test case annotations rather than simple result-set comparison. This enables the precise failure mode diagnosis the reviewer requests.
>
> 2. Full trajectories and conversation logs further reveal planning and communication failures:
>
> Beyond test cases, BIRD-INTERACT records every interaction step: clarification requests, tool invocations, SQL submissions, model thoughts/actions, and debugging attempts. Upon release, these trajectories allow researchers to reconstruct failure chains. We provided an example analysis in Appendix J and Section 5.2. Based on our comprehensive elements for analysis, we can conclude better patterns of interaction of LLMs leading to higher SR solutions: stronger systems follow a clear strategy: early turns combine environment exploration with user clarifications, while mid/later turns increase execute and submit for verification and converting agreed logics to valid SQL outputs.
>
>
> These components jointly provide exactly the multi-level diagnostic visibility for deeper analysis beyond just binary SR results.
>
> [1] Jing, Liqiang, et al. "DSBench: How Far Are Data Science Agents from Becoming Data Science Experts?." The Thirteenth International Conference on Learning Representations. (ICLR' 25)
>
> [2] Lei, Fangyu, et al. "Spider 2.0: Evaluating Language Models on Real-World Enterprise Text-to-SQL Workflows." The Thirteenth International Conference on Learning Representations. (ICLR' 25)

---

> > ### Author Response · Authors · 2025-11-19
> > **Response -- 2**
> >
> > > W3: The rules of the agent game might be pushing models into unnatural strategies. For example, making it computationally "expensive" to ask the user a question could discourage communication and encourage a brute-force, trial-and-error approach. It’s possible the models are just learning how to beat the benchmark’s specific point system rather than learning how to solve database problems effectively.
> >
> > **A3:** We thank the reviewer for this insightful observation, given it touches upon a fundamental challenge in designing agentic benchmarks. The concern that models might learn to "beat the benchmark's specific point system" rather than learn effective problem-solving is a valid one.
> >
> > Our design philosophy for the a-Interact cost structure was to intentionally model the real-world trade-offs that a database assistant would face. In a production environment, both compute cycles and user time are "expensive."
> >
> > Here are narratives and evidences to support this design:
> >
> >  - Reflection from real-world experiences and relevant study: Our decision to make `ask()`  or `submit()` more expensive is based on user study from HCI research ([3], [4]) that note that interrupting users carries real interaction cost and must be carefully timed (Guidelines G3 and G10 in [3]). Excessive clarification questions can impair user experience, so modeling them as costly is consistent with established human–AI interaction principles.
> >
> >  - The higher cost associated with the ask (cost=2) and submit (cost=3) actions is a direct representation for user patience. A system that constantly pesters the user with clarification questions or submits numerous incorrect solutions is not an effective assistant. We want to discourage a "brute-force" communication strategy and instead incentivize the agent to first explore its "cheaper" autonomous options, such as diving the schema (get_schema, cost=1) or knowledge base (get_knowledge_definition, cost=0.5).
> >
> >  - Our empirical results in Section J.2 (Figures 11 and 12) show that the benchmark is successfully differentiating models based on this trade-off:
> >      - We observed that weaker models do fall into the unnatural, "game-playing" strategies the reviewer mentioned. For example, Qwen-3-Coder and Deepseek-Chat-V3.1 adopted a brute-force trial-and-error approach, heavily overusing the execute action (48% and 41% of their actions, respectively). At the other extreme, O3-Mini over-relied on the user, spending 91% of its budget on ask and submit. Both strategies led to poor performance.
> >
> >     - Instead, the strongest performers (GPT-5 and Claude-Sonnet-4) adopted a balanced strategy. They did not abandon the expensive ask action; they simply used it more judiciously after first probing the environment.
> >
> > This evidence suggests that the cost system is working as intended. It is not merely "pushing models into unnatural strategies"; rather, it is rewarding the more "natural" and effective strategy of strategic planning and cost-benefit analysis, while successfully penalizing the less-effective brute-force approaches.
> >
> >  - At last, this is precisely why we also include the c-Interact evaluation setting. This setting removes the agentic "game" and its associated costs, focusing instead on a structured, protocol-guided dialogue. As shown in Table 2, model performance varies significantly between the two modes (e.g., GPT-5 excels in a-Interact but struggles in c-Interact). This confirms that the two settings are indeed testing distinct and complementary skills: c-Interact tests for effective communication within a set protocol, while a-Interact tests for autonomous planning and resource management under constraints.
> >
> > [3] Amershi, Saleema, et al. "Guidelines for human-AI interaction." Proceedings of the 2019 chi conference on human factors in computing systems. (CHI' 19)
> >
> > [4] Kim, Yoonsu, et al. "Understanding users’ dissatisfaction with chatgpt responses: Types, resolving tactics, and the effect of knowledge level." Proceedings of the 29th International Conference on Intelligent User Interfaces. (IUI' 24)

---

### Official Review · Reviewer_nKK2 · 2025-10-27

**Soundness:** 3
**Presentation:** 4
**Contribution:** 3
**Rating:** 8
**Confidence:** 3

**Summary:**

The paper introduces BIRD-INTERACT, a new benchmark designed to evaluate large language models (LLMs) for *interactive* text-to-SQL tasks. Unlike prior static, single-turn datasets (e.g., Spider, BIRD), BIRD-INTERACT simulates realistic multi-turn interactions between a system and a user simulator, capturing the iterative clarification, debugging, and follow-up behavior that occurs in real-world database applications.

**Strengths:**

-  The paper addresses an important gap between single-turn text-to-SQL evaluation and real interactive database usage, providing a benchmark with clear practical relevance.
- The inclusion of knowledge bases, metadata, and executable DBs offers a realistic simulation environment; the two-stage user simulator design is technically sound and well-motivated.
- The paper situates its contribution well within prior work (e.g., COSQL, LEARN-TO-CLARIFY, LIVESQLBENCH) and justifies why static multi-turn datasets are insufficient.

**Weaknesses:**

- Limited discussion of annotation cost and human validation. While expert annotators are mentioned, clearer statistics on time, inter-annotator reliability per ambiguity type, or annotation guidelines would strengthen credibility.
- The experimental setup reveals that each model was executed **only once** because of the *high cost of commercial API calls*, even though the authors used deterministic decoding (temperature=0, top_p=1) to justify single-run reproducibility. While this design ensures consistency, it also raises concerns about the **statistical robustness and variance** of results—especially given the benchmark’s complexity and multi-turn stochastic nature. Moreover, the reported per-model expenses for BIRD-INTERACT-FULL and -LITE (Table 2 and Table 10) suggest that large-scale evaluation is economically prohibitive, potentially hindering broader adoption and replication by the community

**Questions:**

-  A key question is whether performance on *BIRD-INTERACT-LITE* reliably predicts results on *BIRD-INTERACT-FULL*. Quantifying this relationship—such as through rank correlation or regression analysis—could clarify whether the Lite version serves as an efficient proxy for large-scale evaluation, enabling faster model iteration without sacrificing benchmarking fidelity.

---

> ### Author Response · Authors · 2025-11-19
> **Thanks for your Suggestions and Time!**
>
> We appreciate the reviewer’s constructive feedback. Our point-by-point responses are listed below.
>
> > W1: Limited discussion of annotation cost and human validation. While expert annotators are mentioned, clearer statistics on time, inter-annotator reliability per ambiguity type, or annotation guidelines would strengthen credibility.
>
> **A1:** We thank the reviewer's suggestion to further strengthen the credibility of our paper. For more information regarding our annotation process, We provide detailed information below:
>
> **Annotation Cost:** On average, annotators spent 35–40 minutes per task on average to design ambiguities, design follow-up sub-tasks, and verify the execution.
>
> **Inter-Annotator Reliability.** We quantified reliability using a double-blind cross-validation scheme where all the tasks were re-annotated. The overall **Inter-Annotator Agreement (IAA)** is **93.3%** on LITE set and **93.5%** in FULL set shown in Table 1. We further broke down reliability by ambiguity type across two datasets to ensure consistency across different sources of uncertainty: (1) User Query Ambiguity: 92.7%, and (2) Knowledge Ambiguities: 94.3%.
>
> **Annotator Group.** BIRD-INTERACT is developed via a multi-stage annotation process converting clear single-turn tasks into dynamic, interactive scenarios. This involves two annotation groups: 1) **12 qualified database/SQL annotators**, who possess substantial *practical experience in Database operation* and passed a strict entry test (scoring $>90\%$) as detailed in **Appendix C.1** and completed systematic training shown in **Appendix C.2** to promise the quality of annotation; 2) **3 senior database experts/scientists** for final data collection decisions and dispute resolution.
>
> **Annotation & Validation Phase.** Each annotator begins by reproducing single-turn baselines, then systematically design ambiguities (e.g., intent-level, knowledge-based) and constructs follow-up sub-tasks. After annotation, the data undergoes a rigorous **cross-validation** phase, where annotators exchange data for review. This verification involves three steps: (1) executing the generated SQL against the database sandbox to verify functional correctness and state updates; (2) a "red teaming" process where annotators check the necessity and recoverability of the ambiguities; (3) resolving disagreements through discussion, with persistent issues escalated to the expert team for final determination. The inter-agreement scores are reported above.
>
> **Annotation Guideline** can be found in Section 3.2 of our paper with details.
>
> > Q1: A key question is whether performance on BIRD-INTERACT-LITE reliably predicts results on BIRD-INTERACT-FULL. Quantifying this relationship—such as through rank correlation or regression analysis—could clarify whether the Lite version serves as an efficient proxy for large-scale evaluation, enabling faster model iteration without sacrificing benchmarking fidelity.
>
> **A3:** Thanks for your question to make our split more rigorous. Results indicate that *BIRD-INTERACT-LITE is a reliable predictor of BIRD-INTERACT-FULL performance*. The Spearman rank correlations between Lite and Full success rates are high in both interaction mode, where **ρ = 0.8929 (p-value = 0.0068)** for c-interact and **ρ = 0.8571 (p-value = 0.0137)** for a-interact. Because both p-values are far below the standard significance threshold of 0.05, these correlations are statistically significant. This proves your guess, the Lite rankings meaningfully preserve the relative ordering of models observed in the Full benchmark, demonstrating that BIRD-INTERACT-LITE can serve as an effective and efficient proxy for large-scale evaluation.

---

> ### Author Response · Authors · 2025-11-19
> **Response -- 2**
>
> > W2: The experimental setup reveals that each model was executed only once because of the high cost of commercial API calls, even though the authors used deterministic decoding. Moreover, the reported per-model cost much.
>
> **A2:** Thank you for raising this important point. Our design choices were made precisely to ensure deterministic, reproducible, and controllable evaluation despite the high cost of commercial API calls.
>
> 1. Single-run evaluation under deterministic decoding is standard and appropriate in the context of text-to-SQL and other tabular code generation tasks due to long-context nature [1, 2]. To address this, we adopt fully deterministic decoding (temperature=0, top_p=1) to minimize randomness introduced by the model itself as previous accepted works suggested.
>
> 2. The entire evaluation environment is strictly deterministic to reduce randomness:
>
>     - System models use deterministic decoding as we discussed before (temperature=0, top_p=1).
>
>     - The database execution / evaluation environment is reset based on Docker instances for each task, and we set commonly-used PostgreSQL 15 as mainly evaluated version to guarantee identical DB states across reruns.
>
>     - The user simulator is not a free-form generative model. Our two-stage function-driven simulator is more robust as highlighted by your comments and other reviewers.
>
> 3. To further validate the determinism of our setup, we conducted an additional experiment:
> all models were executed five times on a representative subset of tasks.
>
> Table 1. The experiment of running 5 times on c-Interact.
> | Model       | Priority Question BI | Priority Question DM | Priority Question Overall | Follow Ups BI | Follow Ups DM | Follow Ups Overall |
> |-------------|-------------|-------------|------------------|-------------|-------------|------------------|
> | Deepseek-Chat-V3.1     | 11.58 ± 0.41 | 34.60 ± 0.47 | 18.83 ± 0.39     | 4.91 ± 0.40  | 16.72 ± 0.60 | 8.63 ± 0.40      |
> |Qwen-3-Coder-480B | 16.45 ± 0.59 | 35.03 ± 0.69 | 22.30 ± 0.61     | 8.22 ± 0.53  | 17.46 ± 0.37 | 11.13 ± 0.43     |
> | Claude-Sonnet-4    | 16.20 ± 0.47 | 35.77 ± 0.60 | 22.37 ± 0.41     | 10.66 ± 0.32 | 22.43 ± 0.29 | 14.37 ± 0.27     |
>
>
> Table 2. The experiment of running 5 times on a-Interact.
> | Model       | Priority Question BI | Priority Question DM | Priority Question Overall | Follow Ups BI | Follow Ups DM | Follow Ups Overall |
> |-------------|-------------|-------------|------------------|-------------|-------------|------------------|
> | Qwen-3-Coder-480B | 8.32 ± 0.40  | 25.29 ± 0.69 | 13.67 ± 0.31     | 3.80 ± 0.41  | 5.50 ± 0.60  | 4.33 ± 0.37      |
> | Deepseek-Chat-V3.1     | 10.75 ± 0.53 | 31.64 ± 0.78 | 17.33 ± 0.53     | 4.67 ± 0.44  | 5.29 ± 0.53  | 4.87 ± 0.46      |
> | Claude-Sonnet-4| 16.11 ± 0.58 | 55.03 ± 0.84 | 28.37 ± 0.45     | 7.93 ± 0.50  | 23.81 ± 0.84 | 12.93 ± 0.30     |
>
> As we can observed, the standard errors are not much which means our setup is more deterministic and more reproducible.
>
> 4. In fact, the high evaluation cost is an inherent characteristic of database-relevant code generation and text-to-SQL tasks, primarily due to the large and complex database inputs required for each query. This effect becomes even more obvious in multi-turn settings, Nevertheless, we deliberately included evaluations on frontier commercial models because they serve as important purposes:
>     - We include frontier models not to prescribe their use, but to establish an empirical baselines and to diagnose why interactive Text-to-SQL remains challenging even for state-of-the-art systems. Demonstrating that these strongest models still struggle is essential evidence of the value of benchmark and nontriviality which is not a requirement for others to adopt expensive models.
>     - We save costs for the Community. By already conducting extensive evaluations on the most expensive models, our paper absorbs the initial cost of establishing performance baselines.
>     - We will release all interaction trajectories, user-simulator decisions, and execution logs. This provides the community with a rich set of analysis-ready resources for studying frontier model behavior, designing error taxonomies, and developing new evaluation metrics without re-running any expensive experiments.
>
> Finally, we think that your disucssion can bring valuable future research directions for interactive tex-to-SQL tasks. This can encourage the development of more efficient systems that can manage long-context effectively and perform competitively, even surpassing frontier models with lower computational and financial budgets.
>
> [1] Jing, Liqiang, et al. "DSBench: How Far Are Data Science Agents from Becoming Data Science Experts?." The Thirteenth International Conference on Learning Representations. (ICLR' 25)
>
> [2] Lei, Fangyu, et al. "Spider 2.0: Evaluating Language Models on Real-World Enterprise Text-to-SQL Workflows." The Thirteenth International Conference on Learning Representations. (ICLR' 25)

---

### Official Review · Reviewer_h895 · 2025-11-01

**Soundness:** 4
**Presentation:** 3
**Contribution:** 4
**Rating:** 8
**Confidence:** 4

**Summary:**

This paper proposes BIRD-INTERACT, a benchmark for evaluating Text-to-SQL models in realistic, multi-turn interaction settings. It integrates a hierarchical knowledge base, metadata, and an autonomous user simulator to support dynamic clarification and error recovery. The benchmark includes two evaluation modes and full CRUD tasks with injected ambiguities, revealing a significant gap between current LLM capabilities and real-world interactive SQL reasoning.

**Strengths:**

1. The work re-imagines Text-to-SQL evaluation by modeling a full, high-fidelity human-AI collaborative loop, moving beyond the static constraints of previous datasets. The combination of CRUD support, HKB reasoning, and dynamic interaction is unique.
2. The rigorous task annotation (e.g., SQL snippets as clarification sources), the two-stage simulator with proven robustness on USERSIM-GUARD, and the comprehensive experimental methodology attest to the high quality of the benchmark construction.
3. The paper is clearly written, and the analysis provides a deep, multi-faceted understanding of why current models struggle with the interactive paradigm.

**Weaknesses:**

1. Limitation on Simulator Creativity: While the function-driven approach ensures robustness against ground-truth leakage, its reliance on pre-annotated ambiguities and structured retrieval for `LOC()` actions fundamentally limits the model's ability to handle truly novel, unforeseen ambiguities that a real human might introduce. The simulator is robust but sacrifices some degree of open-ended human creativity. Future research should strive to achieve both robustness and broader generalization in the simulation.
2. High Barrier to Entry: The main results rely heavily on expensive, closed-source models (e.g., GPT-5, Claude-Sonnet-4, Gemini-2.5-Pro). The high average cost per task (up to $0.67 for Claude-Sonnet-3.7) creates a significant financial barrier for broad academic community adoption, particularly for large-scale training and iterative development, even with the smaller LITE set. A more prominent emphasis on developing and benchmarking cost-effective, open-source models capable of effective interaction is necessary.

**Questions:**

1. Generalization of the User Simulator (LOC() action): The success of the `LOC()` action depends on retrieving relevant SQL fragments via AST matching. Have the authors considered how their two-stage system would handle an out-of-distribution clarification request where the intent is reasonable, but the answer is not easily localizable to a specific SQL fragment (e.g., asking for an abstract policy not represented directly in the ground-truth SQL)? This would better test the generalizability vs. ground-truth grounding trade-off.
2. Balance of Budget and Efficiency (for a-Interact): The analysis shows models favor costly `submit` and `ask` actions over efficient exploration tools. In designing the action costs, did the authors consider adaptive or dynamic cost schedules? For example, penalizing models that consecutively choose low-value `execute` actions might more strongly incentivize strategic resource exploration and planning.
3. DM Task Distribution: The task suite includes the full CRUD spectrum (DM tasks). To better assess model capabilities in this crucial non-SELECT domain, could the authors provide a breakdown of the distribution of specific DM operations (`INSERT`, `UPDATE`, `DELETE`, `ALTER TABLE`) within the benchmark, along with the performance metrics for models on each of these types? This is vital for evaluating model readiness for operational database tasks.

---

> ### Author Response · Authors · 2025-11-19
> **Thanks for your Suggestions and Questions!**
>
> Thank you very much for the insightful review. We address the raised concerns in the following sections.
>
> > W1 & Q1: The function-driven simulator's reliance on pre-annotated ambiguities and AST-based LOC() retrieval fundamentally limits handling of truly novel, unforeseen ambiguities. How would the system handle out-of-distribution clarifications where answers aren't easily localizable to SQL fragments (e.g., abstract policies)?
>
> **A1:** We thank the reviewer for this thoughtful question about the generalization-robustness tradeoff in simulator design. We clarify that our design explicitly addresses both pre-annotated and unanticipated clarifications while maintaining evaluation fairness:
>
> 1. First, two distinct mechanisms enable both reproducibility and flexibility:
>
>     - AMB() handles pre-annotated ambiguities (ensuring reproducibility)
>     - LOC() specifically handles "reasonable clarification requests that fall outside our pre-annotated ambiguities" as introduced in Section 3.3 and Appendix N. The LOC() function is intentionally designed to support diverse, model-specific clarification strategies that cannot be pre-enumerated. For example:
>
> For instance, in the same task,
>
> GPT-4o ask: *"What specific attributes should be included in the 'profile' for these fans? For example, are you interested in their demographics, membership status, or something else?"*
>
> Claude-Sonnet-4 ask *"What is the exact join condition between the monitoringdevices table and the productbatches table?"*
>
> These questions are task-relevant but difficult to pre-annotate exhaustively. Our `LOC()` mechanism uses semantic AST matching to locate relevant SQL fragments as **referenced intent** and generate contextually appropriate responses, enabling flexible interaction while maintaining grounding in the ground-truth solution.
>
> 2. Benchmark Fairness requires controlled evaluation conditions: As a benchmark (vs. an open-ended dialogue system), maintaining evaluation fairness is important:
> There is a fundamental tradeoff: Allowing unconstrained simulator randomness makes it impossible to determine whether performance differences arise from: (a) System model capabilities, or (b) Random variations in simulator behavior. Our function-driven design ensures:
>
>     - All models receive consistent, appropriate responses to equivalent clarification strategies
>     - Performance differences reflect system capabilities, not simulator noise
>     - Results are more reproducible.
>
> This is also why purely generative baseline simulators achieve only 0.54-0.61 correlation with human users (Table 3), while ours achieves 0.79-0.84 in which the structure enables rather than limits realistic simulation.
>
> 3. LLM-Based Generation (Not Rule-Based Matching) can enable natural responses. We emphasize that annotated SQL snippets serve as **reference context** for LLM generation, not **strict templates**:
> Example: When a model asks "For SAI computation, should I normalize front_run by dividing by 100 to bring it to 0-1 scale like other indicators? Also, how to handle missing or null values in these fields (e.g., set to 0)?"
>
> Locates the relevant SQL fragment: "`0.3 * (COALESCE((ms.manip_signals ->> 'front_run')::real, 0) / 100)`"
>
> Generates a natural explanation via LLM: *"Yes, the \"front_run\" value should be normalized by dividing by 100 to align it with the 0-1 scale of other indicators. For missing or null values, they should be handled by setting them to 0 to ensure the computation proceeds without errors."*
>
> This is not a rule-based system simply returning SQL strings, instead it's LLM-powered generation grounded in ground-truth semantics. This design achieves natural, varied responses while preventing hallucination or ground-truth leakage.
>
> 4. Our design allows researchers to introduce controlled variations as needed. For example, additional knowledge sources (business rules, user profile or styles) can be integrated into `LOC()` without redesigning the architecture.
>
> This modularity is a strength where we provide a reliable foundation while supporting community-driven extensions for specialized research needs.

---

> > ### Author Response · Authors · 2025-11-19
> > **Response--2**
> >
> > > W2: High Barrier to Entry: The main results rely heavily on expensive, closed-source models creates a significant financial barrier for broad academic community adoption, particularly for large-scale training and iterative development, even with the smaller LITE set. A more prominent emphasis on developing and benchmarking cost-effective, open-source models capable of effective interaction is necessary.
> >
> > **A2:** Thank you for bringing this discussion. We illustrate it in following aspects:
> >
> > 1. Our benchmark is fully model-agnostic; researchers can use any model or training paradigm they prefer. We include frontier models (GPT-5, Claude-Sonnet-4, Gemini-2.5-Pro) not to **prescribe** their use, but to establish an empirical baselines and to diagnose why interactive Text-to-SQL remains challenging even for state-of-the-art systems. Demonstrating that these strongest models still struggle is essential evidence of the value of benchmark and challenge. But this is not a requirement for others to adopt expensive models.
> >
> > 2. We save costs for the community by doing these already. By conducting extensive evaluations on the most expensive models, our paper absorbs the initial cost of establishing performance baselines. Researchers can directly compare their local or open-source systems against these published baselines without needing to re-run costly evaluations, effectively reducing their financial burden. And Table 1 & 2 in **A2** to Reviewer nKK2 can prove the performance is robust.
> >
> > 3. We evaluated several open models (e.g., Qwen-3-Coder, DeepSeek-V3, and additional smaller variants). These models still show obvious performance gaps in interactive settings, reinforcing the need for continued research into cost-effective, locally runnable agents due to larger size of these open-source models.
> >
> > Below tables are results of more affordable / deployable models on BIRD-INTERACT-FULL. The success rate is cumulative; Reward* is the normalized reward(%). The values reported in c-Interact are after debugging phase, and (+n) means the performance gained via debugging.
> >
> > Table 1. Performance of entry-level models on c-Interact.
> >
> > | Model       | Priority Question BI | Priority Question DM | Priority Question Overall | Follow Ups BI | Follow Ups DM | Follow Ups Overall | Reward* |
> > |--------------|-------------------------------|--------------|--------------|-------------------------------|--------------|--------------|-------------|
> > | **Llama-3.1-8B**    | 2.92 (+0.73) | 6.35 (+0.00) | 4.00 (+0.50) | 1.22 (+0.97) | 1.59 (+0.53) | 1.33 (+0.83) | 3.02 |
> > | **Qwen3-8B**        | 6.33 (+0.49) | 13.76 (+1.06) | 8.67 (+0.67) | 2.92 (+0.00) | 9.52 (+2.12) | 5.00 (+0.67) | 7.37 |
> > | **Qwen3-32B**       | 6.81 (+1.95) | 13.76 (+1.59) | 9.00 (+1.83) | 5.60 (+0.24) | 6.88 (+1.06) | 6.00 (+0.50) | 7.68 |
> > | **Qwen3-14B**       | 9.49 (+1.46) | 13.76 (+0.53) | 10.83 (+1.17) | 6.08 (+0.00) | 7.94 (+0.53) | 6.67 (+0.17) | 9.33 |
> > | **Llama-3.1-70B**   | 12.41 (+2.43) | 14.81 (+2.12) | 13.17 (+2.33) | 7.30 (+1.22) | 7.41 (+1.59) | 7.33 (+1.33) | 10.82 |
> > | **Qwen3-Coder-30B** | 14.60 (+1.70) | 19.05 (+1.06) | 16.00 (+1.50) | 7.30 (+0.00) | 9.52 (+0.00) | **8.00 (+0.00)** | **13.30** |
> >
> > Table 2. Performance of entry-level models on a-Interact.
> > | Model       | Priority Question BI | Priority Question DM | Priority Question Overall | Follow Ups BI | Follow Ups DM | Follow Ups Overall | Reward* |
> > |--------------|-------------------------------|--------------|--------------|-------------------------------|--------------|--------------|-------------|
> > |  **Llama-3.1-8B**            | 0.25                 | 8.70                 | 2.94                      | 0.25         | 1.63         | 0.69               | 2.27   |
> > | **Qwen3-32B**                | 4.88                 | 24.74                | 11.17                     | 1.71         | 6.32         | 3.17              | 8.77   |
> > | **Llama-3.1-70B**  | 6.10                 | 34.74                | 15.17                     | 2.68         | 7.37         | 4.17              | 11.87  |
> > | **Qwen3-8B**           | 6.83                 | 31.05                | 14.50                     | 2.93         | 8.95         | 4.83              | 11.60  |
> > | **Qwen3-Coder-30B**         | 8.78                 | 38.42                | 18.17                     | 4.39         | 8.95         | 5.83              | **14.47**  |
> > | **Qwen3-14B**           | 5.61                 | 31.58                | 13.83                     | 3.17         | 15.79        | **7.17**              | 11.83  |
> >
> > 4. We think our environment and benchmark supports Accessibility and open research. All benchmark components (schemas, HKBs, simulation rules, interaction scaffolds, evaluation w/ dockers) are uploaded during submission and will be open-sourced.
> >
> > Thanks again, we hope these clarifications can address your concern. Our intention is **not to set a high barrier, but to provide a rigorous and transparent evaluation**.

---

> > > ### Author Response · Authors · 2025-11-19
> > > **Response--3**
> > >
> > > > Q2: Balance of Budget and Efficiency (for a-Interact): The analysis shows models favor costly submit and ask actions over efficient exploration tools. In designing the action costs, did the authors consider adaptive or dynamic cost schedules? For example, penalizing models that consecutively choose low-value execute actions might more strongly incentivize strategic resource exploration and planning.
> > >
> > > **A3:** Thank you for the insightful question regarding adaptive or dynamic cost schedules. Our design choice for a-Interact was guided by the goal of creating a stationary and comparable evaluation environment.
> > >
> > > 1. Fixed Costs Enable Controlled, Interpretable Analysis & Comparison: We adopted a fixed cost schedule to ensure that agent behaviors are comparable across models and runs. A stationary cost function allows us to attribute differences in performance directly to the model’s ability rather than variability introduced by shifting reward structures.
> > >
> > > 2. However, we think your suggestion is highly valuable as a method for training or optimizing interaction policies. As discussed in Section J, we observe that many models behave greedily toward `ASK` and `SUBMIT`. Your proposed dynamic penalties could indeed discourage repetitive or low-value actions and encourage agents to explore tools more strategically, effectively pushing them from greedy behavior toward more e-greedy or adaptive policies.
> > > We view this as a promising direction of method development built on top of our benchmark.
> > >
> > > Thank you for raising this interesting point that highlights how our benchmark can reveal and encourage such policy innovations, ultimately benefiting both the Text-to-SQL and open-source agent research communities.
> > >
> > > > Q3: DM Task Distribution: The task suite includes the full CRUD spectrum (DM tasks). To better assess model capabilities in this crucial non-SELECT domain, could the authors provide a breakdown of the distribution of specific DM operations (INSERT, UPDATE, DELETE, ALTER TABLE) within the benchmark, along with the performance metrics for models on each of these types? This is vital for evaluating model readiness for operational database tasks.
> > >
> > > **A4:** We appreciate the reviewer’s question and have added breakdown of distributions. The benchmark includes the following distribution of DM operations:.
> > >
> > > | DM Operation | Statistics (%) |
> > > |-------------|----------------|
> > > | UPDATE      | 45.07        |
> > > | INSERT      | 28.17         |
> > > | ALTER TABLE | 14.09          |
> > > | DELETE      | 12.68          |
> > >
> > >
> > > We also report model performance on Bird-Interact-Full separately for each DM operation: (We select 3 popular models here due to limited space.)
> > >
> > > ## (1) c-interact Performance by DM Operation
> > >
> > > ### Gemini-2.5-Pro
> > > | CRUD Type | Priority Question SR (%) | Follow Ups SR (%) | Avg Reward |
> > > |-----------|-----------|-----------|------------|
> > > | INSERT | 29.63 | 25.93 | 24.44 |
> > > | UPDATE | 59.09 | 34.09 | 47.95 |
> > > | DELETE | 77.78 | 66.67 | 68.89 |
> > > | ALTER TABLE | 26.67 | 13.33 | 22.67 |
> > >
> > > ### Claude-Sonnet-4
> > > | CRUD Type | Priority Question SR (%) | Follow Ups SR (%) | Avg Reward |
> > > |-----------|-----------|-----------|------------|
> > > | INSERT | 25.93 | 14.81 | 19.63 |
> > > | UPDATE | 56.82 | 34.09 | 47.27 |
> > > | DELETE | 66.67 | 55.56 | 55.56 |
> > > | ALTER TABLE | 33.33 | 33.33 | 32.67 |
> > >
> > > ### GPT-5
> > > | CRUD Type | Priority Question SR (%) | Follow Ups SR (%) | Avg Reward |
> > > |-----------|-----------|-----------|------------|
> > > | INSERT | 11.11 | 7.41 | 9.26 |
> > > | UPDATE | 40.91 | 27.27 | 36.14 |
> > > | DELETE | 44.44 | 11.11 | 34.44 |
> > > | ALTER TABLE | 20.00 | 13.33 | 18.00 |
> > >
> > >
> > >
> > > ## (2) a-interact Performance by DM Operation
> > >
> > > ### Claude-Sonnet-4
> > > | CRUD Type | Priority Question SR (%) | Follow Ups SR (%) | Avg Reward |
> > > |-----------|-----------|-----------|------------|
> > > | ALTER TABLE | 60.00 | 33.33 | 52.00 |
> > > | DELETE | 88.89 | 44.44 | 75.56 |
> > > | INSERT | 22.22 | 7.41 | 17.78 |
> > > | UPDATE | 72.73 | 36.36 | 61.82 |
> > >
> > > ### Gemini-2.5-Pro
> > > | CRUD Type | Priority Question SR (%) | Follow Ups SR (%) | Avg Reward |
> > > |-----------|-----------|-----------|------------|
> > > | ALTER TABLE | 60.00 | 33.33 | 52.00 |
> > > | DELETE | 66.67 | 44.44 | 60.00 |
> > > | INSERT | 25.93 | 7.41 | 20.37 |
> > > | UPDATE | 40.91 | 18.18 | 34.09 |
> > >
> > > ### GPT-5
> > > | CRUD Type | Priority Question SR (%) | Follow Ups SR (%) | Avg Reward |
> > > |-----------|-----------|-----------|------------|
> > > | ALTER TABLE | 66.67 | 46.67 | 60.67 |
> > > | DELETE | 100.00 | 66.67 | 90.00 |
> > > | INSERT | 29.63 | 14.81 | 25.19 |
> > > | UPDATE | 75.00 | 38.64 | 64.09 |
> > >
> > > Thanks for asking this, which makes our experiment presentation more interpretable and comprehensive.

---

### Official Review · Reviewer_4t8u · 2025-11-01

**Soundness:** 3
**Presentation:** 3
**Contribution:** 3
**Rating:** 8
**Confidence:** 4

**Summary:**

This paper introduces BIRD-INTERACT, a new benchmark for Text-to-SQL designed to address the limitations of existing static, single-turn, and read-only datasets. It provides a dynamic, multi-turn evaluation framework where models must handle ambiguity, execution errors, and evolving user goals. The benchmark's core components include a task suite covering the full CRUD spectrum, an interactive environment with a knowledge base, and a novel function-driven user simulator that allows models to ask clarification questions without human supervision. The paper introduces two evaluation settings: c-Interact (a structured, protocol-guided conversation) and a-Interact (an open-ended agentic setting). Empirical results demonstrate the benchmark's significant difficulty, with top-tier models like GPT-5 achieving success rates below 18%, highlighting a substantial gap between current LLM capabilities and the demands of real-world dynamic database tasks.

**Strengths:**

- **S1. Addresses a Critical Research Gap:** The paper correctly identifies a major limitation in existing Text-to-SQL research. The field has largely focused on single-turn SELECT query generation, which does not reflect the ambiguous, iterative, and stateful nature of real-world data analysis. This benchmark shifts the focus to a more realistic and challenging interactive paradigm.
- **S2. Robust User Simulator:** The two-stage, function-driven user simulator is a significant contribution. By first mapping a model's question to a symbolic action (AMB, LOC, UNA) before generating a response, it avoids the common pitfalls of naive LLM-based simulators, such as ground-truth leakage and un-controlled-generation.
- **S3. Comprehensive and Realistic Task Design:** The benchmark's design is high-quality. It expands the task scope beyond SELECT to include the full CRUD spectrum (DML/DDL), which is a crucial step towards evaluating real-world database assistants. It also introduces state dependency, where follow-up sub-tasks depend on the state changes (e.g., a newly created table) from a previous sub-task. Finally, the principled injection of ambiguities (in queries, knowledge, and the environment) makes interaction a necessity for success.

**Weaknesses:**

- **W1. Unclear Knowledge Ambiguity Taxonomy:** The distinction between "one-shot knowledge ambiguity" and "knowledge chain breaking" in Section 3.2 is unconvincing. The paper's own example for "knowledge chain breaking" is to mask the intermediate node "AVS". However, masking "AVS" is functionally identical to removing it as a "one-shot" ambiguity (defined as removing an "isolated knowledge entry"). Both actions result in the "AVS" node being missing. This conceptual overlap makes the taxonomy feel redundant and fails to delineate a meaningful difference between the two ambiguity types.
- **W2. Questionable Premise of "Implementation-Level Ambiguity":** The paper categorizes user under-specification as "implementation-level ambiguity" (Section 3.2 and Appendix H.4), which seems to be a conceptual flaw. For example, the query "Show recent purchases" is presented as ambiguous, with the "clarified" version being "Show recent purchases sorted by time". A query that is merely less specific is not inherently ambiguous, as any result set of recent purchases, regardless of sort order, would be a correct answer. Similarly, "Show average score" is treated as ambiguous, with the "clarified" version being "Show average score in 2 decimal". The un-rounded, full-precision answer is also correct. This design choice seems to conflate ambiguity with a lack of specific formatting instructions, risking unfair penalization of models that make reasonable default assumptions.

**Questions:**

- **Q1. On "Implementation-Level Ambiguity":** Could the authors elaborate on the reasoning for classifying under-specification as "ambiguity"? Using the example from Appendix H.4, "Show recent purchases" is a valid query. Why is a model penalized for not clarifying a specific sort order (ORDER BY purchase_time DESC) that was never requested? Does this not create a benchmark that tests for a specific, "over-specified" ground truth rather than functional correctness?
- **Q2. On "Decimal Ambiguity":** Following up on Q1, why is a query like "Show average score" considered ambiguous? Is the full-precision answer (e.g., 85.12345) considered incorrect? It seems that requesting rounding (e.g., ROUND(AVG(score), 2)) is an additional formatting instruction, and its absence doesn't make the original query ambiguous.
- **Q3. On "Knowledge Ambiguity" Distinction:** Could you please clarify the distinction between "one-shot knowledge ambiguity" and "knowledge chain breaking"? The paper's example for chain breaking is to mask the 'AVS' node. How is this functionally different from removing the 'AVS' node under the "one-shot" definition? Both actions seem to result in the same missing node. Could you provide an example where these two ambiguity types are mutually exclusive and force different system behaviors?

---

> ### Author Response · Authors · 2025-11-19
> **Thanks for your Reviews!**
>
> We are grateful for the reviewer’s detailed comments and have addressed each point below.
>
> > W1 & Q3: W1. The distinction between "one-shot knowledge ambiguity" and "knowledge chain breaking" in Section 3.2 need clarify the conceptual difference and provide a concrete example demonstrating where these two ambiguity types are mutually exclusive and force distinct system behaviors.
>
> **A1:** Thanks for asking this! The distinction between "one-shot knowledge ambiguity" and "knowledge chain breaking" is based on the (relational) **knowledge dependency** of the masked knowledge within the Hierarchical Knowledge Base (HKB).
>
> * **One-shot knowledge Ambiguity:** This type will mask an isolated knowledge that has no other dependencies. Take Figure 2 as an example with user query: `Calculate and rank all artifacts by their Conservation Priority Index (CPI).`
> In HKB, the knowledge `CPI --> (10 - conservation status) / 30` and `conservation status` can be directly mapped to the database column `conservation_status`. When we mask this knowledge item, it creates an ambiguity that requires clarification, therefore the term "one-shot knowledge ambiguity" means resolving it requires only a single piece of information.
>
> * **Knowledge Chain Breaking:** This type will only happens when the knowledge required for precisely produce SQLs for such user query is multi-step with each connecting each others. Also take Figure 2 as an example, if user query is: `Build a function to calculate and rank all artifacts to identify which need urgent care.` and the knowledge path underneath `need urgent care` is `urgent care` $\rightarrow$ `AVS` $\rightarrow$ `IF` & `CPI`, If we mask the intermediate node `AVS`, the chain is severed. Unlike the one-shot scenario, the system cannot simply ask for the final formula `(IF/CPI)` because it does not yet know that `AVS` depends on `(IF/CPI)`.
>
> Thus masking "AVS" **ONLY** belongs to the Knowledge Chain Breaking type and is **NOT** functionally identical to removing it as a "one-shot" ambiguity.
>
> Also these two types lead to different failure modes shown in our detailed analysis. In Section K, we can observe system models can solve one-shot ambiguities once the definition is supplied mechanically . In contrast, "Chain Breaking" requires the model to perform deep fault localization which should identify where in the reasoning path the link is broken. Our results show that most models struggle significantly more with Chain Breaking because they fail to infer the existence of the missing intermediate link, whereas One-shot ambiguity is treated as a simple retrieval task. These experiments can provide empirical validation for our categorization of such two ambiguity types and support the research definitions we adopt in this work.

---

> > ### Author Response · Authors · 2025-11-19
> > **Remaining Response**
> >
> > > W2 & Q1 & Q2: The classification of user under-specification as "implementation-level ambiguity" appears conceptually flawed, as it conflates a lack of specific formatting instructions. This creates a critical concern that the benchmark tests for arbitrary, over-specified ground truths rather than functional correctness, potentially unfairly penalizing models for failing to predict unrequested constraints.
> >
> > **A2** Thanks for raising this point! We want to clarify that our inclusion of implementation-level ambiguity is reflected from real-world requirements, but, **crucially**, our evaluation metric is explicitly designed to avoid the unfair penalization you maybe concerned about.
> >
> > **1. Why implementation-level ambiguities are very important?**
> >
> > The classification of certain ambiguities as "implementation-level" is from empirical findings on large-scale crowdsourcing in prior benchmark research [1], which identified that most evaluation disagreements stem from implementation choices such as `DISTINCT` usage. Drawing on these findings along with our further extensive consultation with database experts, we systematically catalogued common implementation variations that prove that leads to huge difference in practice. Therefore, we annotate these features (shown in details in Table 7) to create a benchmark that reflects production-grade standards rather than simplified academic tasks.
> >
> > **2. Addressing the "Unfair Penalization" Concern: We Use Soft Exact-Match Evaluation**
> >
> > To directly address the concern about "testing for arbitrary ground truths," our default implementation actually took into account. As detailed in Appendix G, the default evaluation uses a **Soft Exact-Match (Soft-EM)** mechanism that prevents models from being penalized for failing to predict unrequested constraints.
> >
> > Before comparison, the test script systematically normalizes predictions by removing benign differences such as `remove_comments`, `remove_distinct`, and `remove_round`. This ensures that unless a specific constraint is explicitly part of the functional requirement (and thus clarified during interaction), the model is judged on the core logic of the SQL.
> >
> > In summary, implementation-level ambiguity annotations exist to allow for **optional**, **fine-grained** analysis of a model's ability to align with specific business rules. For example, in financial reporting, strict decimal precision (e.g., ROUND(x, 2)) is mandatory. In data migration or auditing, specific join logic (e.g., LEFT JOIN vs. INNER JOIN) determines data integrity.
> >
> > However, our primary metric remains functional correctness, and researchers can simply switch on these constrains into consideration by following instructions in our submitted code.
> >
> > [1] Li, Jinyang, et al. "Can llm already serve as a database interface? a big bench for large-scale database grounded text-to-sqls." Advances in Neural Information Processing Systems 36 (2023): 42330-42357. (NeurIPS' 23)

---

### Author Response · Authors · 2025-11-19
**General Response**

We would like to sincerely thank all reviewers for their time and helpful feedback. We feel encouraged that all reviewers found the benchmark **"addresses an important research gap"** and acknowledge our user simulator's **"robustness"** and **"technical soundness"** (Reviewer 4t8u, gEbn, nKK2, h895). Reviewers also think our benchmark is a **"huge and necessary step forward"** (Reviewer gEbn) that **"re-imagines Text-to-SQL evaluation"** (Reviewer h895); the design of our user simulator is **"technically sound"** (Reviewer nKK2) and **"is incredibly smart"** (Reviewer 4t8u). Thanks for your appreciation in our work.

Our detailed responses answer cover the specific topics below to strengthen credibility:

1.  **Why our User Simulator will not lose creativity** (Reviewers h895, gEbn): We have clarified the special design of our user simulator in detail, where controlable randomness is allowed.
2.  **High cost of commercial API calls actually provide benefits for benchmark** (Reviewers nKK2, h895): We provide results on low-cost open-sourced LLMs and prove the benchmark results are stable and robust.
3.  **Further clarification on quality of our benchmark** (Reviewers nKK2, 4t8u): We further clarify the process of human annotation & validation process in detail and justify for our ambiguity classification.

We carefully answer all concerns below and we are happy to answer any further questions.

---

### Meta-Review · Area_Chair_4Lh4 · 2026-01-05

**Summary:**

**1) Summary**
This paper introduces BIRD-INTERACT, a benchmark aimed at evaluating Text-to-SQL systems in realistic, multi-turn, ambiguity-rich database interaction settings. It features full CRUD task coverage, a dynamic environment, and a function-driven user simulator designed to support clarification, error recovery, and state-dependent reasoning. Experiments show that even top commercial LLMs struggle, revealing a substantial capability gap between current Text-to-SQL technology and real-world interactive workflows.

**2) Strengths**

* Introduces a benchmark that addresses a major gap by moving from single-turn SQL generation to realistic, multi-turn, stateful database interactions.
* Presents a robust, carefully engineered two-stage user simulator that avoids ground-truth leakage and supports structured handling of ambiguity.
* Offers comprehensive task design with CRUD coverage, knowledge bases, metadata, and principled ambiguity injections that reflect real-world use cases.
* Provides clear empirical evidence that current state-of-the-art models underperform substantially, highlighting important open challenges in interactive Text-to-SQL reasoning.

**3) Weaknesses**

* Certain ambiguity taxonomies and definitions (e.g., implementation-level ambiguity, distinctions among ambiguity types) appear conceptually weak or insufficiently justified.
* The benchmark’s reliance on expensive commercial APIs and high per-task costs presents adoption barriers, and limited discussion of annotation cost and statistical robustness further constrains replicability.

**Reviewer Concerns:**

The reviewers had consistent opinions in favor of acceptance, and the strengths of the paper significantly outweigh its weaknesses.

**Reviewer Scores:**

The reviewers did not seem to change their scores.

---

### Decision · Program_Chairs · 2026-01-26

Accept (Oral)